# A method to estimate the cellular composition of the mouse brain from heterogeneous datasets

**Dimitri Rodarie**◉*, **Csaba Verasztó**◉, **Yann Roussel**◉, **Michael Reimann**◉, **Daniel Keller,**
**Srikanth Ramaswamy, Henry Markram, Marc-Oliver Gewaltig**◉

Blue Brain Project, École polytechnique fédérale de Lausanne (EPFL), Geneva, Switzerland

* dimitri.rodarie@epfl.ch

## Abstract

The mouse brain contains a rich diversity of inhibitory neuron types that have been characterized by their patterns of gene expression. However, it is still unclear how these cell types are distributed across the mouse brain. We developed a computational method to estimate the densities of different inhibitory neuron types across the mouse brain. Our method allows the unbiased integration of diverse and disparate datasets into one framework to predict inhibitory neuron densities for uncharted brain regions. We constrained our estimates based on previously computed brain-wide neuron densities, gene expression data from *in situ* hybridization image stacks together with a wide range of values reported in the literature. Using constrained optimization, we derived coherent estimates of cell densities for the different inhibitory neuron types. We estimate that 20.3% of all neurons in the mouse brain are inhibitory. Among all inhibitory neurons, 18% predominantly express parvalbumin (PV), 16% express somatostatin (SST), 3% express vasoactive intestinal peptide (VIP), and the remainder 63% belong to the residual GABAergic population. We find that our density estimations improve as more literature values are integrated. Our pipeline is extensible, allowing new cell types or data to be integrated as they become available. The data, algorithms, software, and results of our pipeline are publicly available and update the Blue Brain Cell Atlas. This work therefore leverages the research community to collectively converge on the numbers of each cell type in each brain region.

## Author summary

Obtaining a global understanding of the cellular composition of the brain is a very complex task, not only because of the great variability that exists between reports of similar counts but also because of the numerous brain regions and cell types that make up the brain.

Previously, we presented a model of a cell atlas, which provided an estimate of the densities of neurons, glia and their subtypes for each region in the mouse brain. Here, we describe an extension of this model to include more inhibitory neuron types. We collected estimates of inhibitory neuron counts from literature and built a framework to combine

**Data Availability Statement:** All relevant data are within the manuscript and its Supporting Information files. The code is available at:https://github.com/BlueBrain/atlas-densities Results can

be visualized at: https://bbp.epfl.ch/nexus/cell-atlas/ The data downloaded from the Allen Institute website is available at the following links (see also Acknowledgements Section in the main text): Nissl CCFv2: http://download.alleninstitute.org/informatics-archive/current-release/mouse_ccf/ara_nissl/ Annotation atlas CCFv2: http://download.alleninstitute.org/informatics-archive/current-release/mouse_ccf/annotation/mouse_2011/ Annotation atlas CCFv3: http://download.alleninstitute.org/informatics-archive/current-release/mouse_ccf/annotation/ccf_2017/ The AIBS ISH dataset can be found at http://mouse.brain-map.org/search/index The corresponding experiment identifiers are listed below: • Parvalbumin: Experiment id #868 used for the BBCAv2 and for S8 Fig • Somatostatin: #1001 for the BBCAv2 • Vasoactive Intestinal Peptide: #77371835 for the BBCAv2 • GAD1 (equivalent of GAD67): #479 for the BBCAv2, #79556706 for Fig 3, #480 for Fig 10, #75457536 for S7 Fig and #79556706 used in Section 3.4 • GAD2 (equivalent of GAD65): #79903740 for S7 Fig.

**Funding:** This study was supported by funding to the Blue Brain Project, a research center of the École polytechnique fédérale de Lausanne (EPFL), from the Swiss government's ETH Board of the Swiss Federal Institutes of Technology. The funders had no role in study design, data collection and analysis, decision to publish, or preparation of the manuscript.

**Competing interests:** The authors have declared that no competing interests exist.

them into a consistent cell atlas. Using brain slice images, we also estimated inhibitory neuron density in regions where no literature data are available. We estimated that in the mouse brain 20.3% of all neurons are inhibitory. Among all inhibitory neurons, 18% predominantly express parvalbumin (PV), 16% express somatostatin (SST), 3% express vasoactive intestinal peptide (VIP), and the remainder 63% belong to the residual GABAergic population. Our approach can be further extended to any other cell type and provides a resource to build tissue-level models of the rodent brain.

This is a *PLOS Computational Biology* Methods paper.

## 1. Introduction

Over the past century, numerous studies have reported a great variety of cells in the mouse brain according to their morphological, electrical, and molecular features. Several groups such as the US BRAIN Initiative have launched ambitious projects to undertake a comprehensive census of all brain cells, determining their molecular, structural and functional properties [1–4]. Building a catalog of the vast diversity of neuron types in the brain is a challenge for modern neuroscience. Characterizing this variety is exacerbated by the fact that distinct cell types are localized to specific brain regions—for example, pyramidal cells in the cerebral cortex or Purkinje cells in the cerebellum [5,6]. Therefore, studies tend to simplify this task by focusing on specific brain regions or a small subset of known cell types. For example, there is a small number of studies that attempt a brain wide estimation of cell densities, using molecular and microscopic techniques to label cells across all brain areas, however, these studies are always limited to only a few cell classes [7–10]. However, these brain-wide datasets cover only a small proportion of the mouse brain cell classifications. Moreover, the variety of methods used has resulted in considerable variability in the outcomes, even when the same region is considered [7,11,12]. As a result, it is difficult for neuroscientists to evaluate the quality of published data sets and to combine different data sets (even brain-wide mappings) to estimate the cellular composition of the entire brain. Compiling a complete and comprehensive cell atlas of the mouse brain is, therefore, a monumental task, which needs to be tackled to enable the *in silico* reconstruction of multiscale neural circuits [6,13–16].

In this paper, we provide a method to generate a cell atlas model of the whole mouse brain. This method integrates disparate datasets from literature for multiple cell types, using a constrained optimization approach. The final cell densities in our model are constrained by one another to maintain a coherent framework.

### 1.1. First version of the mouse brain cell atlas

Erö and colleagues [17] were able to estimate the cellular composition of each brain region defined by the Allen Mouse Brain Atlas [18] with respect to excitatory (glutamatergic) and inhibitory (GABAergic) neurons, astrocytes, oligodendrocytes, and microglia. The authors aligned multiple datasets, including genetic marker expression from *in situ* hybridization (ISH, see complete list of abbreviations in Table A in S1 Document) experiments from the Allen Institute for Brain Science (AIBS), to the Allen Reference Nissl volume from Lein et al. [19] to determine the positions of all cells within the annotation volume. This resulted in the

first version of the Blue Brain Mouse Cell Atlas pipeline (BBCAv1) and the associated model as described in Erö et al. [17].

Erö et al. argued that literature values are often inconsistent and are therefore unreliable [12,17]. Hence, the authors used only whole-brain values from literature (e.g., the total number of cells and neurons in the mouse brain from Herculano-Houzel et al. [20]) to constrain their estimates of cell densities in every brain region, avoiding any bias toward a particular region. However, there are also several regions for which a consensus on cell composition has been reached (e.g., for the cerebellar Purkinje layer, see review from Keller et al. [12]). For instance, in all regions of the isocortex (also called neocortex), it is well-known that layer 1 regions should only be composed of inhibitory neurons [5]. To take this aspect into account, the BBCAv1 had to manually integrate those regional constraints *a posteriori* (e.g., force each neuron in isocortex L1 to be inhibitory) which in turn contradicted the original global constraints. Moreover, the BBCAv1 strategy yielded regional estimates that sometimes did not match their respective literature values (e.g. cell estimates for cortical subregions compared to Herculano-Houzel et al. [21]). In order to obtain a more fine-grained classification, the BBCAv1 strategy required brain-wide estimates for every new cell type to be integrated. These constraints may not yet exist. We therefore decided to refine the BBCAv1 strategy to obtain inhibitory neuron subtypes densities.

## 1.2. A novel approach to estimate cell densities

We extended the workflow from Erö et al. [17], using estimates of cell densities for individual brain regions from the literature instead of whole-brain (global) estimates for specific cell types.

Specifically, using additional ISH experiment datasets from the AIBS, we were able to further refine the GABAergic neuron densities from Erö et al. [17] and estimate the densities of GABAergic subclasses. This includes parvalbumin (PV+), somatostatin (SST+), vasoactive intestinal peptide (VIP+) immunoreactive neurons, and the remaining inhibitory neurons.

We chose to include these subclasses in the new pipeline because they have been extensively studied in the literature [7,8,22–52], which means we could find data to build and validate our model. Moreover, the associations of these neuron types with brain health or disease make them an important refinement to our model. PV+ neurons are fast spiking, resulting in fast and efficient suppression of their surrounding neurons' activity [53,54]. A deficit of PV+ neurons seems to be linked to neural diseases such as schizophrenia [42], Alzheimer's [26] and autism [32,36]. SST+ neurons contribute to long range inhibition [54]. Their population decreases with age and seems to contribute to the development of Alzheimer's disease [39,41,47]. VIP+ neurons, on the other hand, tend to target other inhibitory neurons in the isocortex, playing the role of local disinhibitors [38]. SST+ and VIP+ neurons are also involved in the circadian clock which regulates daily brain activity [55]. As in Erö et al. [17], we did not use counting algorithms to extract counts of cells from the AIBS ISH datasets. Indeed, the authors pointed out that counting techniques were not always producing satisfactory results especially in dense regions because of cell overlapping (see their Figs 1C and 7A–7D). Using counting techniques at the whole brain scale is therefore out of the scope of this study.

We also highlighted the fact that the alignment of multiple datasets to a common coordinate system is essential to obtain the best results in terms of cell density estimations. We therefore used Krepl et al.'s algorithm [56], based on deep learning techniques to automatically realign ISH datasets to our reference system. This tool replaces the manual realignment step in Erö et al.'s pipeline [17]. We combined all these new tools and methods to produce a second version of the Blue Brain Mouse Cell Atlas pipeline (BBCAv2) and model. We further

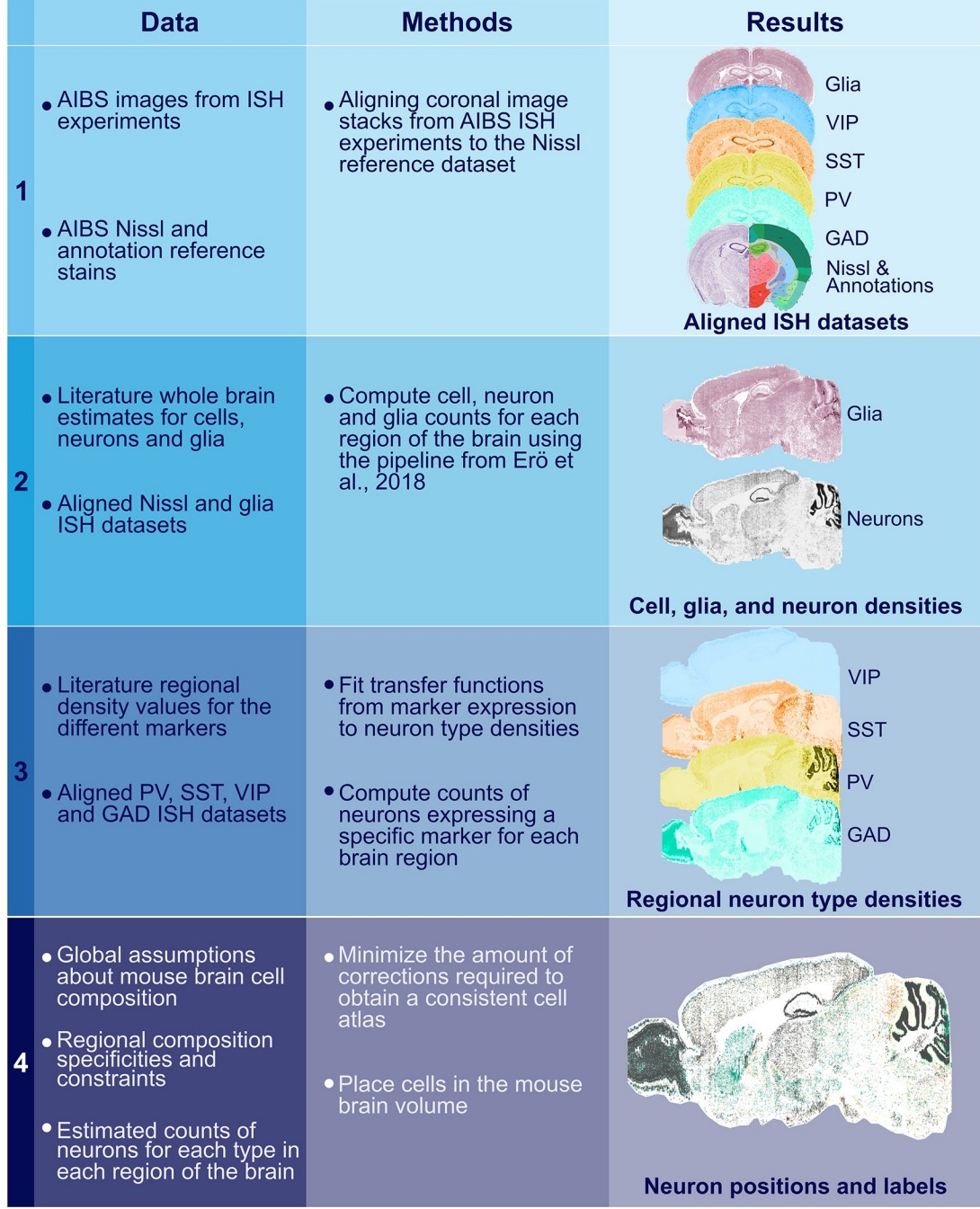

**Fig 1. New workflow for the Blue Brain Mouse Cell Atlas (BBCAv2).** Steps of the pipeline are displayed in rows with the data consumed in the left column, the method used in the middle column, and the produced results on the right. Each step of this pipeline builds on the results of the previous steps.

extended the BBCAv2 in a companion paper [57], mapping well identified morpho-electrical types to our GABAergic neuron subclasses. These refinements to the model will help neuroscientists to get a better understanding of cellular composition in the brain and will pave the way for more accurate *in silico* reconstructions of brain tissues.

## 2. Materials and methods

### 2.1. Overview of the pipeline

**2.1.1. Original pipeline.** In this paper we refined several steps of the BBCAv1 to facilitate the integration of more and more specific cell types without having to rely on the total number or ratio of these cell types. We started with the original Blue Brain Mouse Cell Atlas workflow (BBCAv1) from Erö et al. [17] which consisted of six steps:

1. The Nissl-stained brain slices from Dong [18], and the corresponding mouse brain Annotation Volume (AV) were manually realigned along the sagittal axis. Every ISH dataset [19] used in the following steps was also manually realigned to match the Nissl volume.

2. The annotated Nissl volume from the AIBS was combined with a total number of cells in the mouse brain from Herculano-Houzel et al. [20] to estimate the number of cells in each brain region.

3. A combination of genetic marker datasets and a global ratio of glial cells in the brain from Herculano-Houzel et al. [20] was used to distinguish neurons from glial cells.

4. Glial cells were labeled as astrocytes, oligodendrocytes or microglia based on whole brain ratios together with their respective ISH datasets.

5. GAD67 (glutamic acid decarboxylase) marker experiment from Lein et al. [19], associated with a global number of inhibitory neurons from Kim et al. [7] was used to distinguish the excitatory from the inhibitory neurons. This number stands for the sum of PV+, SST+ and VIP+ reacting cells in the brain and was assumed in this pipeline to represent the entire inhibitory population.

6. Manual correction of purely inhibitory regions: molecular layer regions of the cerebellum, layer 1 of the isocortex, the reticular nucleus of the thalamus, and the striatum were considered to be fully inhibitory, which means that all their neurons are labeled as GABAergic.

**2.1.2. Adjustments for improved pipeline.** Step 1 of the BBCAv1 involved manual selection of landmark points on each slice of the AV, and ISH data to realign to the Nissl volume. This step was not only labor intensive but also error prone. In our updated workflow, we replaced the manual realignment by a novel deep learning based alignment algorithm by Krepl et al. [56].

In step 5, the total number of inhibitory neurons in the mouse brain was a global constraint. Its biological value was taken from Kim et al. [7] and represents the sum of SST+, PV+, and VIP+ neurons. However, these three types do not represent all inhibitory neurons (e.g., LAMP5 cells in the isocortex [58]). This means that the total number of inhibitory cells in BBCAv1 was underestimated, despite the corrections applied in step 6.

We therefore reworked steps 5 and 6 by integrating more literature estimates of neuron type densities as well as by using additional ISH datasets. This not only improved our estimates of the excitatory to inhibitory ratios in the brain, but also provided more precise estimates of GABAergic neuron types, including PV+, SST+, and VIP+ cells. However, as we will discuss later, the spatial resolution (200 μm between each coronal slice) and the whole-brain coverage of these datasets are not sufficient to estimate the density of all inhibitory subtypes in each voxel of the mouse brain, as it was done in BBCAv1. Therefore, we computed the mean density of each inhibitory type for each region of the AV, instead of estimating the density per voxel.

Since several AV and Nissl volumes have been released [59,60], we also developed a method to choose the best combination to estimate cell densities (see Section 2.2).

**2.1.3. New pipeline.** Fig 1 shows the new pipeline for the Blue Brain Mouse Cell Atlas version 2 (BBCAv2). It consumes data from the AIBS in the form of image stacks of stained coronal brain slices, as well as a combination of annotation and Nissl reference atlases.

The BBCAv2 has four main steps.

1. In the first step, the different image stacks from ISH experiments from the AIBS are automatically aligned and registered to a reference volume. We describe this step in Section 2.3.

2. The second step generates cell, neuron, and glial densities for each region of the AV. It corresponds to steps 2 and 3 of the BBCAv1 pipeline.

3. We devise assumptions to estimate densities of different inhibitory neuron types in a coherent framework (see Section 2.4) and apply them for step 3 and 4. In step 3, we assume a correlation between genetic marker expression and cell type density gathered from literature (see Section 2.5) and the previously realigned and filtered ISH datasets. The details of this step are described in Section 2.6.

4. Step 4 integrates the results of step 3 into a consistent estimate of the cell densities for each inhibitory subtype, according to our assumptions (see Section 2.7). Finally, the cells are placed in the brain volume to produce the updated BBCAv2 model (see Section 2.8).

## 2.2. Selection of the reference brain atlas based on the resulting cell density distribution

Step 2 of the BBCAv2 (see Fig 1) uses a whole-brain volume, derived from Nissl-stained coronal brain slices from Dong [18], together with an annotation volume (AV) which maps the voxels of the brain volume to a brain region. Brain regions are organized in a hierarchical tree, with leaves representing the finest subdivisions captured by the AIBS. These reference volumes are registered by the AIBS to a common vector space: the Common Coordinate Framework (CCF).

The AIBS has released multiple versions of the Allen Brain Nissl volume and the related annotation volumes.

The first version (Nissl1 + AV1) was published in 2011 and was based on a single mouse. Nissl1 + AV1 were well aligned to each other at the level of individual slices, however adjacent slices were not. This led to rather a jagged Nissl brain volume and corresponding annotations. Therefore Erö et al. [17] realigned adjacent Nissl and AV slices along the sagittal axis with a non-rigid algorithm to create a new CCF (CCFbbp).

The AIBS used a comparable technique on the same dataset to obtain Nissl2 + AV2 (CCFv2). It still represents one individual, but this time aligned to an averaged mouse brain volume that was derived from more than a thousand adeno-associated virus (AAV) tracer experiments [61,62]. There are two major differences between CCFv2 and CCFbbp. First, the annotation volume used in Erö et al. (old version of the AV2) does not include labeling of ventricular systems, whereas CCFv2 does. Second, the method for realigning and combining the Nissl images into a single volume is not the same.

A third AV version (AV3) was completely derived from this average brain, using the pipeline described in Wang et al. [60]. AV3 is currently the smoothest AV, but it is no longer derived and aligned to a Nissl volume (see Fig 2A); it is paired with the closest Nissl volume available (Nissl2). Moreover, some subregions such as the molecular and granular layers of the cerebellum were not re-drawn in the AV3.

Clearly, for the purpose of estimating cell densities, different combinations of Nissl volumes and annotation atlases yield different cell density estimates (see Fig 2C and 2D). In this section,

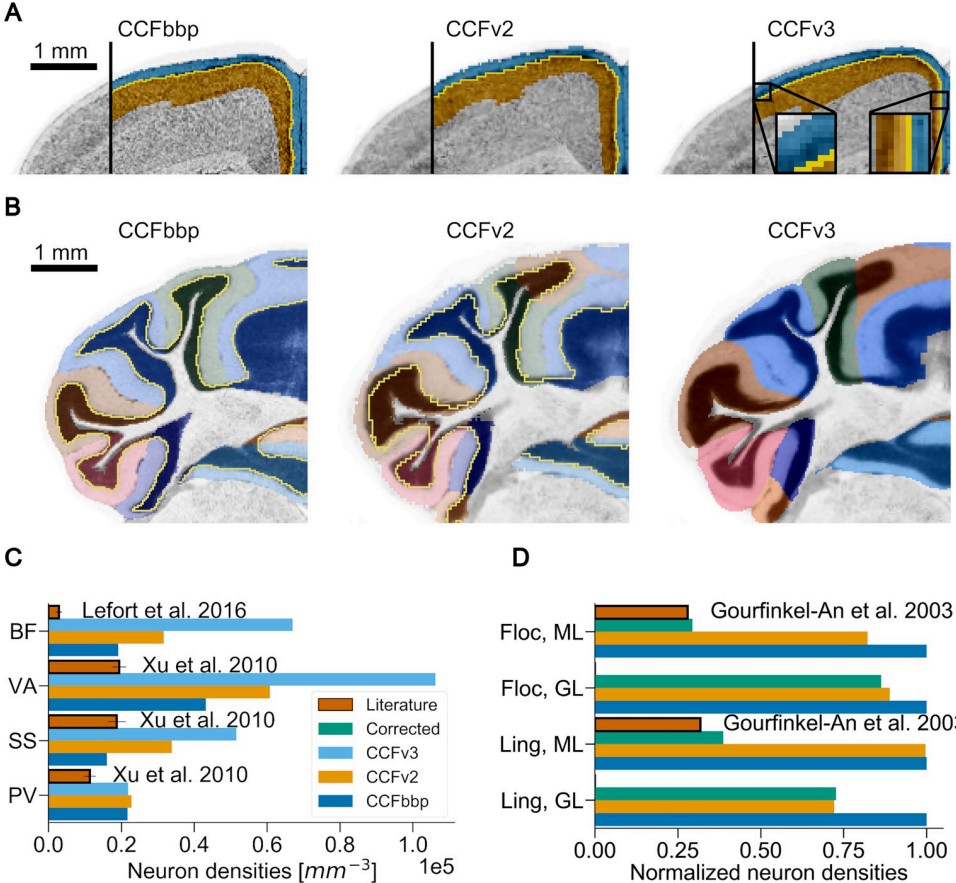

**Fig 2. Local impact of the annotation atlas on the estimation of cell densities.** Panels (A) and (B) show the misalignment between the Nissl datasets in grey and their respective annotation atlases overlaid as colors (coronal slices). Boundaries between annotated regions, when available, are displayed in yellow. Different versions are shown from left to right: CCFbbp, CCFv2 from the AIBS, and CCFv3 from the AIBS. (C) and (D) compare the estimated neuron densities from the BBCAv2 pipeline (step 1 and 2 of Fig 1) based on each reference atlas version to literature (bars in brick red with black outline). Each reference atlas version appears in different colors: dark blue for CCFbbp, orange for CCFv2, light blue for CCFv3, and green for the manually realigned AV2 and Nissl2. (A) Focus on the isocortex L1/L2 boundary. On the right of each image, L1 and L2 annotations are shown respectively in blue and orange, on top of the Nissl expression. The density difference between L1 and L2 creates a visible boundary (see left part of the images). However, in each CCF version, parts of the denser L2 are annotated as L1, raising the estimated density of neurons for L1. The magnifications shown for CCFv3 highlight some examples of this problem. (B) Focus on the cerebellar cortex, granular/molecular boundary (Purkinje layer not represented). Each subregion of the cerebellar cortex in the annotation atlas is displayed on top of the Nissl dataset with a different color. Molecular (ML) and granular (GL) layers are shown in different shades of the same color, except for CCFv3 which does not distinguish these layers anymore. Here also, the annotation boundaries do not follow the visible density changes between these layers in the Nissl data. (C) Focus on various regions of the isocortex L1. For these regions, the neuron counts produced with the pipeline using each CCF version are greater than their literature counterpart. The errors linked to misalignment will increase with the boundary surface area. Using the CCFv3 version (light blue bars) yields the worst estimates. Regions shown: Primary somatosensory area, barrel field (BF); Primary visual area (VA); Primary somatosensory area (SS); Prelimbic area (PV). (D) Focus on lingula (Ling) and flocculus (Floc) regions of the cerebellum. Results of the manual alignment of the AV2 with Nissl2 are shown with green bars. Density estimates are normalized according to the densities obtained with the CCFbbp. We observe that after manual realignment, estimates tend to be closer to literature values than for the other CCF versions. CCFv3 estimates are not shown, because its AV lacks the separation between molecular and granular layers.

we therefore present our analysis to identify the best reference dataset pair (referred to as CCF version) for density estimation. We show that the quality of cell density estimates is strongly correlated to the alignment of the Nissl volume to the AV.

Poor alignment means that parts of one region in the Nissl volume are mislabeled in the AV as being part of a neighboring region. This means also that parts of the cell estimates (step 2 of Fig 1) are assigned to wrong regions. To assess the quality of the alignment we look at parts of the brain where the boundary between two regions is clearly visible in the Nissl volume, specifically the isocortex L1/L2 separation and the cerebellar cortex molecular/granular layers. Fig 2A and 2B illustrates the misalignment between Nissl and AV volumes for the different reference atlases for these regions.

We apply the first two steps of our pipeline (see Fig 1), for the three pairs of annotation/Nissl volumes:

- the CCFbbp version derived from an old version of the Niss2 and AV2. which we name Nissl2a and AV2a.

- CCFv2 version (Nissl2 + AV2) from the AIBS also derived from the Nissl1 and AV1. These datasets are publicly available on the AIBS website [63,64].

- CCFv3 version (Nissl2 + AV3) from the AIBS [65], 2017 release version (most recent).

For each pair of reference volumes, we first estimate the densities of cells and neurons per voxel in the corresponding brain volume, based on the total number of neurons from Herculano-Houzel et al. [20]. We then compare the resulting distribution of densities to each other and to the literature (see Fig 2C and 2D) [51,66,67]. Despite an overall agreement (see S1 Fig), we observe important differences for smaller regions. For regions with low cell density such as isocortex L1, this difference can scale up to two times more cells for CCFv3 as compared to CCFv2 (see Fig 2C), and even more compared to some literature values. To quantify more precisely the impact of such misalignments, we manually realign AV2 to Nissl2 for the lingula and flocculus regions of the cerebellum. We find that this indeed leads to density estimates closer to their literature counterparts (see Fig 2D).

We reproduce the Fig 7A from Erö et al. [17], comparing the literature values against the corresponding estimates, using the different pairs of reference atlases. However, the results are not conclusive enough to select a reference atlas version because most of the generated cell counts agree with their literature counterparts within their range of variability (see S1 Fig).

Our earlier observations indicated that misalignments lead to overestimation of densities for low density regions and to underestimation in high density regions, reducing the differences between regions and thus, making their values more homogeneous. We quantify this by calculating the standard deviation of the neuron count distribution for each CCF. We find that the standard deviation is the largest for the version of CCFbbp (CCFbbp $2.82\times10^6$, CCFv2 $2.26\times10^6$, CCFv3 $2.57\times10^5$) and therefore least affected by misalignment. We therefore adopt CCFbbp throughout the rest of this paper.

## 2.3. Alignment of *in situ* hybridization coronal brain slices

For step 3 of the BBCAv2, we consume ISH datasets from the AIBS [19] which are sectioned in the coronal plane (one 25 μm thick section every 200 μm, see Fig 3D). However, the individual images are not aligned to the same Nissl volume that we use to estimate neuron densities. In the BBCAv1, ISH images slices were manually realigned to the Nissl volume with a non-rigid landmark-based algorithm (see S8 Fig). This task was very tedious and time consuming because it required an expert to manually choose common landmarks between adjacent sections. For our new pipeline, we therefore replace this step with a fully automatic machine learning based method. The results of the automated realignment on a single slice are shown in S8 Fig.

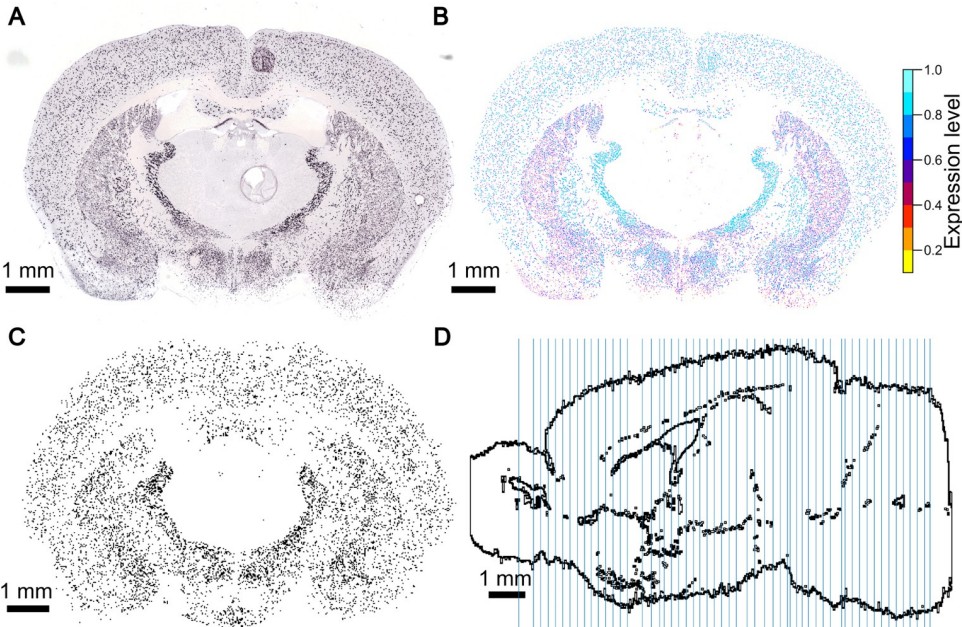

**Fig 3. AIBS GAD67 ISH experiment data.** (A) raw image, (B) filtered image and (C) realigned binarized image. This AIBS ISH experiment highlights inhibitory cells of the mouse brain (experiment No 79556706—slice 22). The raw slice image (A) contains artifacts and a minimum expression value offsets the dataset. The filtered image (B) is derived from the raw image (A) by the AIBS. Somas reacting to GAD67 are detected in the raw image and isolated from the background. The different colors of the cells in (B) represent the different levels of expression (AIBS 2015 white paper). (A) and (B) have been downloaded on the AIBS website [68] with permission of the AIBS. The filtered images are realigned and thresholded to obtain the binarized image shown in (C). (D) Positions of the manually realigned coronal stained slices, shown in blue on a sagittal slice of the CCFbbp brain volume.

The details of this automatic alignment method are described in a separate paper [56].

The different steps of the automatic realignment of every ISH dataset from the AIBS to the reference atlas are as follows:

- Original ISH images (or raw images—see Fig 3A) and their filtered versions, later called filtered images (see Fig 3B), are downloaded from the AIBS website. The filtered images isolate the somas reacting to their specific marker from the brain background in the raw image.

- Each image is scaled, positioned, and rotated according to the reference brain volume, using the provided metadata. The metadata includes the position of the corners of the images because the slices are not perfectly vertical. Here we assumed the slices to be vertical as the algorithm was trained with this assumption and it makes the interpolation between slices easier. Hence, we used the mean position of the coronal ISH images in the AIBS average brain along the rostral-caudal axis also from the metadata. The ISH brain slices, the AV2 and the Nissl2 volume have been all registered within the average brain [62]. So, the ISH coronal section ids should correspond to the Nissl2's. For the Missl1a, used here, for each ISH coronal slice to realign, we manually selected the best corresponding reference Nissl slice.

- For each dataset, every coronal section is automatically realigned to the corresponding anatomical section of the Nissl-stained mouse brain using the Krepl algorithm [56]. Here, the raw *in situ* images are used as they provide more landmarks for the algorithm.

- Resulting displacement fields are then applied to the filtered images of the corresponding ISH experiment.

- A 3D volume for each gene is created by downsampling the images to match the final voxel dimensions of the reference atlas. Missing sections between the 2D coronal slices are generated with a linear intensity-based interpolation.

The resulting 3D voxel-based datasets are consumed by our pipeline to estimate inhibitory neuron subtype densities (see Section 2.6 and S8 Fig).

## 2.4. Assumptions underlying the estimation of inhibitory subtypes densities

We wish to obtain the number of inhibitory neurons and their subtypes for each region of the AV. The regions are indexed by region ids $\mathbf{R} = \{0, r_1, r_2, \ldots, r_n\} \in |N^{861}$, with 0 denoting the outside of the mouse brain. We are particularly interested in inhibitory subtypes expressing the markers GAD67, PV, SST and VIP.

First, the number of neurons can be expressed as the sum of the different subtypes in each region r, $\forall\, r \in \mathbf{R}$:

$$nNeu_r = nInh_r + nExc_r + nOther_r \tag{1}$$

where nNeu, nInh, and nExc are respectively the total, the inhibitory, and the excitatory neuron numbers. As reported by Zeisel et al. [69], these populations seem to be mutually exclusive. The term nOther represents the number of neurons that are neither excitatory nor inhibitory, meaning they could be purely modulatory neurons.

The number nInh can be estimated by the number of neurons, reacting to the markers GAD67 and GAD65 [69] which implies that nInh = nGAD.

We further subdivide nGAD into inhibitory subtypes positively reacting to the markers PV, SST and VIP. This yields the following sum, $\forall\, r \in \mathbf{R}$:

$$nGAD_r = nPV_r + nSST_r + nVIP_r + nRest_r \tag{2}$$

Here nPV, nSST and nVIP are respectively the number of PV+, SST+ and VIP+ immunoreactive neurons in the brain region r. The term nRest corresponds to the number of inhibitory neurons which react neither to PV, SST nor to VIP (InhR neurons), including, for instance, the LAMP5 cells in the isocortex [58]. InhR neurons might define various populations of GABAergic cells, reacting to a great variety of markers [69], which makes it more difficult to estimate nRest. It is therefore easier to estimate nGAD from literature and the marker GAD67 (for regions not covered by literature). Similarly, nPV, nSST and nVIP can be estimated from literature and the markers PV, SST and VIP, respectively. Finally, we can estimate nRest by subtraction in Eq (2). The remaining neuronal populations (nExc + nOther) can be deduced from nGAD (Eq (1)) and the neuron distribution (nNeu) obtained by step 2 of our BBCAv2 pipeline (see Fig 1).

In this estimation, we rely on the following assumptions:

1. Every GABAergic neuron expresses GAD67 and every GAD67 reacting cell is a GABAergic neuron. This genetic marker is indeed responsible for over 90% of the synthesis of GABA [70,71]. Additionally, no cells expressing GAD65 without expressing GAD67 have been reported in the RNA-sequence study from Zeisel et al. [69].

2. GAD67, PV, SST and VIP are only expressed in neurons [55,69,72,73].

3. PV+, SST+ and VIP+ populations are non-overlapping i.e., there are no cells in the mouse brain that co-express a combination of these markers. This assumption is supported in the isocortex and hippocampus by transcriptomic studies such as Huang and Paul [58] and Zeisel et al. [69] and we extrapolate these findings to other areas.

4. Every PV+, SST+ and VIP+ neuron also expresses GAD67 as observed by Celio [74] and Tasic et al. [75].

5. Neuronal composition is homogeneous within subregions at the lowest level of the AIBS region hierarchy.

We will also consider the cell, glia and neuron density distributions obtained in step 2 of the pipeline (see Fig 1) to be correct. These numbers have been validated against literature by Erö et al. [17] and can be refined in the future.

Based on these assumptions, BBCAv2 will provide estimates of the densities of GAD67+, PV+, SST+, and VIP+ neurons for each region of the brain. In contrast with BBCAv1, this new version will present estimates that are as close as possible to available literature values.

## 2.5. Literature review of estimates of the mouse inhibitory neuron densities

For the wild-type adult mouse, the AV2a defines 861 regions. Ideally, we would need at least one literature value for the cell density of each marker in each of these regions. We should also have multiple estimates of the same quantity to increase the reliability of the estimation. The extensive review of Keller et al. [12] provided us with references for inhibitory neuron numbers in the mouse brain (e.g., [30,34,40,42,43]). Among them, the extensive dataset of Kim et al. [7] gives a complete measurement of the distribution of the PV+, SST+ and VIP+ cells in the mouse brain, covering 97% of the regions of our selected AV. We also made a comprehensive literature search for cell densities or cell counts of GAD67, PV, SST, and VIP immunoreactive neurons in the mouse brain (wild-type species).

In total, we found 33 references providing values for GAD67 [5,6,24,28,30,34,35,38,40,45,46,48,49,51,66,67,76–91], 29 for PV [7,22–28,30–37,40–52], 14 for SST [7,8,22,27,31,33,34,36,39,45–47,49,51] and 9 for VIP [7,22,27,29,31,38,45,49,51], extracted from 54 different papers (see S1 Document and Table 1). These estimates are in the form of density of neurons or cell counts within either a volume or a slice. For counts with no volume provided, we used the corresponding volume within the AV to estimate a mean density within the respective region. Finally, some literature sources report percentages of labelled neurons or cells within a region volume. For these estimates, we used the distributions obtained in step 2 of our pipeline (see Fig 1).

Erö et al. [17] considered the neurons of some regions to be inhibitory only (step 6 of the BBCAv1 workflow). Similarly, we imposed this rule on layer 1 of the isocortex, the molecular

**Table 1. Summary of papers reviewed by marker.**

| | Number of papers | Number of regions covered | Coefficient of variation | |
|---|---|---|---|---|
| | | (out of 861) | Median CV | CV Skew. |
| GAD67 | 33 | 151 (17.5%) | 0.20 | 1.19 |
| PV | 29 | 836 (97.1%) | 0.24 | 2.94 |
| SST | 14 | 849 (98.6%) | 0.13 | 3.74 |
| VIP | 9 | 849 (98.6%) | 0.22 | 1.90 |

The number of papers from which we extract density values as input for our model are listed in the first column. The table also shows how many regions of the brain are covered by these values. For each region where we obtain an estimate from at least two different papers, we can compute a coefficient of variation. From the list of these coefficients of variation, we then extract the median values and the skewness of the distribution, defined as: $s = \frac{1}{n} \sum_{r \in R} \left[ \frac{CV_r - mean(CV_r)}{std(CV_r)} \right]^3$ which are displayed for each marker in the last two columns.

layer of the cerebellar cortex and the reticular nucleus of the thalamus. This means that for these regions, we assume that the density of GAD67+ neurons corresponds to its neuron density. Thus, in total we collected GAD67+ density estimates for about 17.5% of all regions of the AV (see Fig 4A).

For some regions such as the CA1 field of the hippocampus, we observe a large variability of the literature estimates (see Fig 4B): spanning from 317 GAD67+ cells per mm$^3$ computed from Han et al. [81] to 7166 GAD67+ cells per mm$^3$ according to Jinno et al. [77] (see the complete list of literature data in the review S1 Excel). In addition to the inter-subject variability, these large variations can arise from a combination of different factors, such as differences in the staining methods, the counting methods, or the methods to determine subregion boundaries (annotations) [7,11]. In the Han et al. and Calfa et al. [81,88] papers, the mean density values extracted were respectively very low and very high when compared to all other literature mean values in the same regions (at least 5 times smaller/bigger). These two papers were therefore not considered for the following steps of our pipeline, since their finding was conflicting.

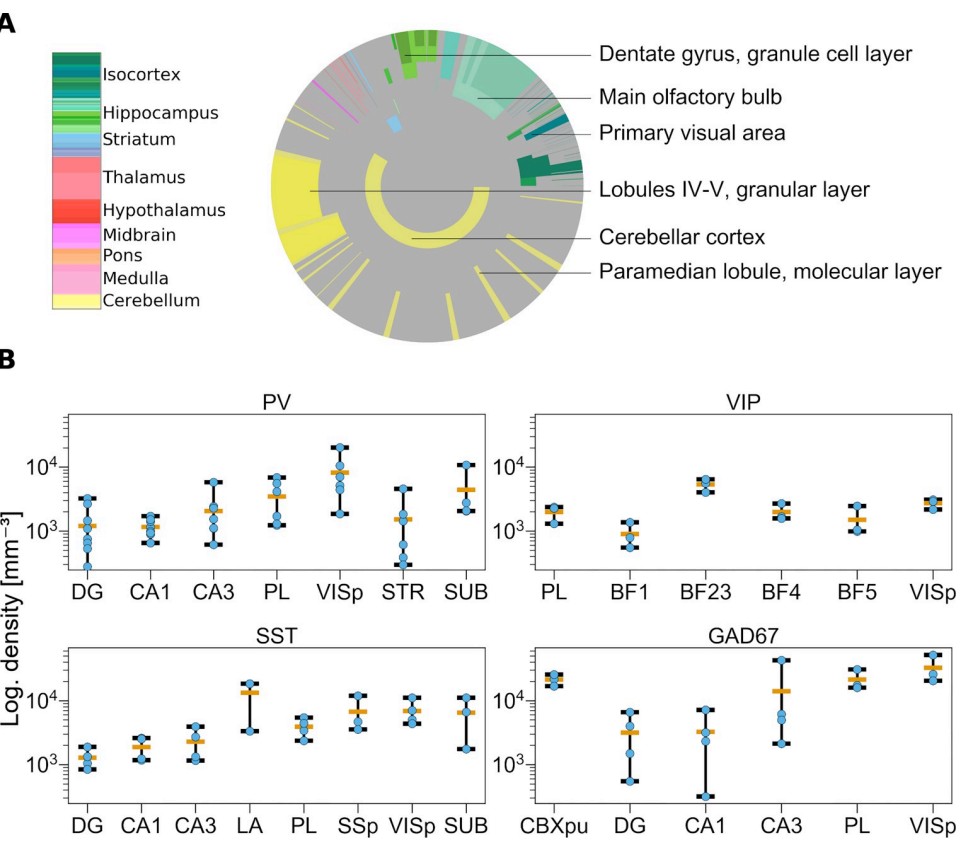

**Fig 4. Overview of published data on inhibitory neuron densities in the mouse brain.** (A) Illustration of regions for which information on densities or absolute numbers of GAD67+ neurons were published. The disk is divided into rings and sectors. Each ring represents a hierarchical level in the AV and sectors represent the contained regions. The center of the disk represents the entire brain, each surrounding ring then represents the next hierarchy level. Colored areas represent regions where at least one study reports cell numbers or densities. The size of the sectors corresponds to the number of neurons in the region relative to the number of neurons in the brain. The colors correspond to regions according to the AV, such as the cerebellar cortex in yellow and cortical areas in green. (B) Variability of published cell density estimates. Each panel shows the range of reported cell densities (log scale). For each region, data points are depicted in blue, the average in orange. The minimum and maximum values are indicated by the whiskers. Regions shown: Dentate Gyrus (DG); Field CA1 (CA1); Field CA3 (CA3); PreLimbic area (PL); Primary SomatoSensory area (SSp); Primary VISual area (VISp); STRiatum (STR); SUBiculum (SUB); Primary somatosensory area, barrel field (BF); Lateral Amygdalar nucleus (LA); CereBellar corteX, Purkinje layer (CBXpu).

Our literature review highlights the current lack of knowledge and consensus on cell composition in the mouse brain, emphasized by the large variability of published data. In the following section, we will describe a method to estimate cell densities in regions for which no literature values were published.

## 2.6. Transfer functions from marker expression to cell density

We use the realigned ISH datasets obtained in Section 2.3 to estimate the densities of inhibitory cell types in regions where no literature values were reported.

Different cell types express genetic markers at different levels [69]. Nonetheless, we want to obtain counts for every cell expressing the markers, no matter their expression level. We do this by applying an all-or-none filter to the realigned filtered image stacks (Fig 3B) using Otsu's method [92] to define the threshold. This forces each pixel showing a cell reacting to the marker to the value 1 and all other pixels to 0 (binarized image stacks see Fig 3C). The more pixels in a region are set to 1, the more densely the region is filled with marker-positive neurons. We therefore expect, for a specific brain region, that its mean pixel intensity (region mean intensity) correlates with the density of marker positive cells. This correlation is further supported when computing the Pearson correlation coefficient ($\rho = 0.59$) between the extracted mean region intensities and corresponding literature density values. This correlation is prevalent across genetic markers (see S9A Fig). S9A Fig shows indeed a clear relationship between literature values and region mean intensity. We also observe a wide spread of the points, especially for low values).

As we have seen above there is a large variability in the cell densities reported by literature (see Fig 4B). This explains the spreading of the points along the y axis in S9A Fig (different point colors). Similarly, points along the x axis are also spread out because of the sparseness of the ISH datasets (see Fig 3D) and its misalignment to the AV (see Fig 2).

We use this relationship to construct transfer functions linking densities of GAD67+, PV+, SST+, and VIP+ cells reported in the literature to the mean intensity in each region of the AV. We leverage every literature density value available for each subregion of the brain, assigning the same weight to each point. Logically, we expect that there is a monotonically increasing relationship between region mean intensity and cell type density. The simplest assumption is that these relations are linear and can be calculated by fitting (black line in S9A Fig). Then, to evaluate the quality of the linear fit from region mean intensity to densities of cells, we compute the coefficient of determination $R^2$ for our linear function f, which fits the cloud of points (x, y) where x is the region mean intensity and y is the literature density:

$$R^2 = \sum_{i \in x}(f(i) - \bar{y})^2 / \left(\sum_{i \in x}(f(i) - \bar{y})^2 + \sum_{i \in x}(y(i) - f(i))^2\right) \tag{3}$$

We obtained a coefficient of determination $R^2 = 34.8\%$, which indicates that the great variability of (x, y) could not be fully accounted for. We therefore tried to improve the quality of our fitting, computing separately a transfer function for each of our genetic markers (see S9B Fig, different colors indicate the different markers). We observe that each population seems indeed to spread differently (see also alpha values obtained with this linear fit in S2 Excel), which supports our choice of splitting according to markers.

With this split, the $R^2$ value improved for PV (50.5%), and VIP (45.6%), stagnated for SST (34.7%), but decreased for GAD67 (16.8%). This drop of fitting quality for GAD67 can be explained by the fewer number of literature values available for the fitting of GAD67 when compared to the other markers. It can also be linked to the great variability in soma sizes or morphologies within the different regions of the brain. Hence, to further improve our transfer

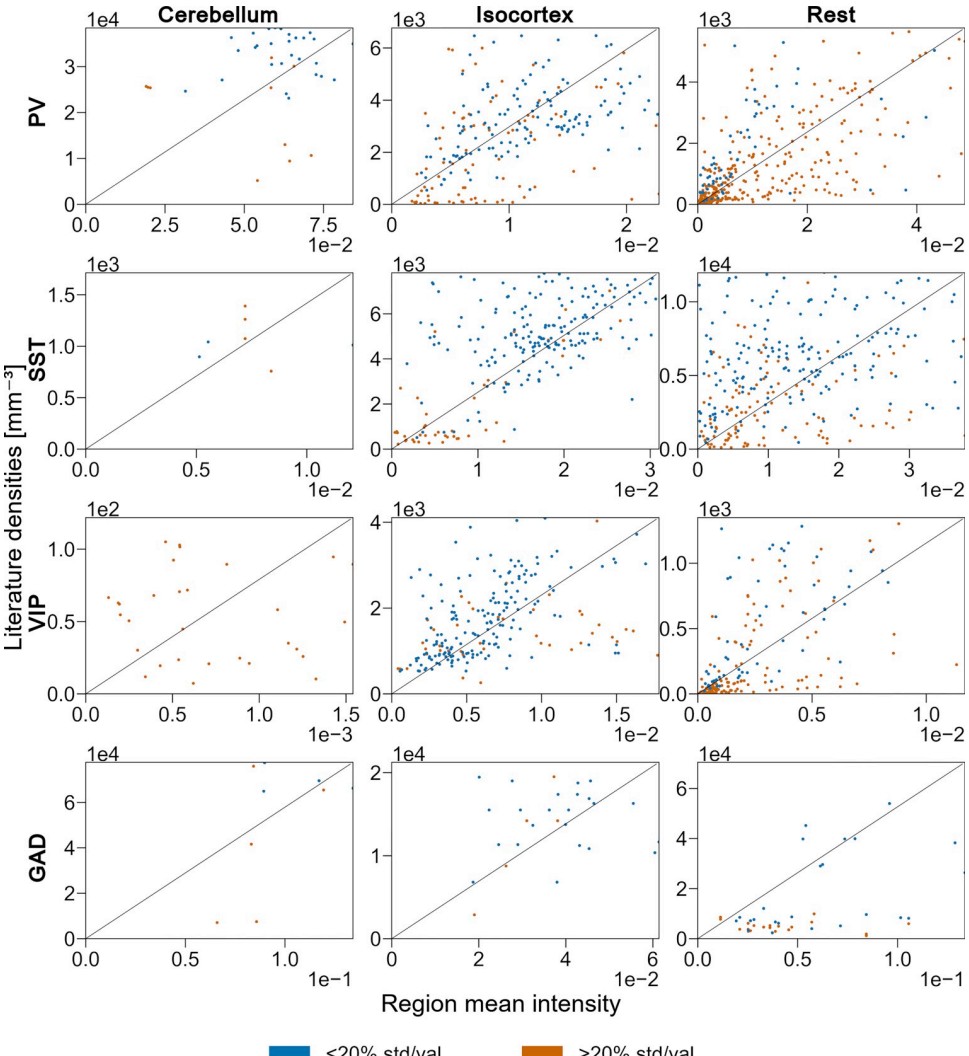

**Fig 5. Linear fitting of marker intensity to cell density in cerebellum, isocortex, and the rest of the brain.** Scatter plots of the PV+, SST+, VIP+ and GAD+ densities reported in literature (y-axis) according to the region mean intensity (x-axis) grouped by the main regions of the brain, respectively cerebellum, isocortex, and the rest of the brain. Each point represents a single literature density value and is color-coded according to the different levels of confidence (ratio of standard deviation over mean value) from literature data. The linear fit is represented with a black diagonal line. Each panel shows 95% of the available points. Regions that are purely inhibitory (i.e., based solely on the neuron counts obtained at step 2 of the pipeline), with no inhibitory neurons and regions with null region mean intensity are not displayed and were not considered for the fitting.

functions, we compare separately the cerebellum, isocortex and the rest of the brain. We choose this division because in the cerebellum, cell densities are very high compared to the rest of the brain [20]. And, inhibitory cell densities and cell type composition across layers of the isocortex are similar in all of its subregions [7]. The results of the comparison between region mean pixel intensity and cell type density are shown in Fig 5.

We expect a region with no inhibitory neurons to have a null mean intensity and conversely a region with null mean intensity should not host any inhibitory neurons. However, we do not observe this property in many cases. This observation can be linked, on the one hand, to the misalignment between the annotation atlas and the ISH datasets (cells reacting appear in the wrong region) or on the other hand, to the insufficient coverage of the brain by the ISH data

(region of interest does not appear on the ISH slices, see Fig 3D). Therefore, these points are excluded from Fig 5. Additionally, as we discussed in Section 2.2, the density estimation in fully inhibitory regions tends to be highly impacted by misalignment. The points linked to these regions are therefore also excluded from the figure. At the first glance, there seems to be no clear relationship.

At the first glance, the relationship seemed to be less clear than the previous linear fits with all data (see S9 Fig). The values of $R^2$ for GAD67 and SST however did improve with this region split (roughly 26% of the variability accounted for GAD67, and 39% for SST see Table 2). The $R^2$ for PV decreased slightly (46%) but is quite constant across the different groups. Finally for VIP, the quality of the fit decreased with this new split. This can be explained in Cerebellum, as literature reports none or very few VIP cells [7,69] in these regions which means that the variability is quite important there. For the isocortex, this lower $R^2$ value might indicate a greater morphological variability in VIP expressing neuron types, such as the ones reported in Huang and Paul [58].

Thus, we use the fitted relation only to predict densities of GAD67+, PV+, SST+ and VIP + in regions where no literature data are available. The standard deviations of these estimates are derived from the standard deviation of the alpha value of the fit (see Table 2 and S2 Excel).

For regions covered by the literature we instead use the average of all means and standard deviations of the reported values (see previous section). We assume indeed that the different literature values scatter around a common average.

Therefore, at this stage we have obtained unconstrained estimates of density ($\eta$) and their standard deviation ($\sigma$) for PV+, SST+, VIP+ and GA67+ neurons for all regions, i.e. $\forall\, r \in \mathbf{R}$:

$$\eta_r = \{nPV_r; nSST_r; nVIP_r; nGAD_r\} \qquad (4)$$

$$\sigma_r = \{std(nPV_r); std(nSST_r); std(nVIP_r); std(nGAD_r)\} \qquad (5)$$

$\eta$ and $\sigma$ are lists of 3444 values: one per neuron type (PV, SST, VIP, GAD67) and per region of the brain (861 in AV1a).

## 2.7. Combination of neuron type densities

Since our unconstrained estimates of inhibitory neuron ($\eta$ and $\sigma$) densities are derived from the transfer functions or independent literature sources, their combination can lead to incongruent results. For instance, the unconstrained estimates of PV+ neurons might be greater than the estimates of GAD67+ neurons, which contradicts our assumption 4 (see Section 2.4). We therefore want to ensure that $\eta$ and $\sigma$ match our assumptions from Section 2.4. To this end, we deduce a list of linear constraints based on Eqs (1) and (2), $\forall\, r \in \mathbf{R}$:

$$0 \leq nPV_r \leq nNeu_r \qquad (6)$$

**Table 2. Results of the linear fitting.**

|  | Cerebellum | | Isocortex | | Rest | |
| --- | --- | --- | --- | --- | --- | --- |
|  | **alpha** | **$R^2$** | **alpha** | **$R^2$** | **alpha** | **$R^2$** |
| PV | $4.557 \times 10^5$ | 43.1% | $2.964 \times 10^5$ | 46.4% | $1.182 \times 10^5$ | 47.9% |
| SST | $1.417 \times 10^5$ | 33.0% | $2.523 \times 10^5$ | 50.6% | $3.158 \times 10^5$ | 34.3% |
| VIP | $7.903 \times 10^4$ | 26.4% | $2.311 \times 10^5$ | 37.4% | $1.154 \times 10^4$ | 56.5% |
| GAD | $5.783 \times 10^5$ | 27.6% | $3.458 \times 10^5$ | 37.0% | $5.265 \times 10^5$ | 13.5% |

The slope factor, alpha, of the linear fitting, and its coefficient of determination $R^2$ for each major region of the brain and for each genetic marker.

and similarly for $nSST_r$, $nVIP_r$ and $nGAD_r$.

$$-nNeu_r \leq nPV_r + nSST_r + nVIP_r - nGAD_r \leq 0 \qquad (7)$$

We also ensure that the consistency of the region hierarchy is maintained for every parent region ($R_m \in \mathbf{R}$). The number of inhibitory neurons (and their subtypes) in $R_m$ is equal to the sum of the corresponding estimates of $R_m$'s direct child regions in the region hierarchy (children$_{Rm}$) plus the estimates in voxels of the AV labeled as belonging to $R_m$ but none of its children ($R_m$\child):

$$nPV_{Rm\ child} + \sum\nolimits_{r \in \text{children}_{Rm}} nPV_r = nPV_{Rm} \qquad (8)$$

and similarly for $nSST_{Rm}$, $nVIP_{Rm}$ and $nGAD_{Rm}$.

Thus, whenever one of the constraints (6), (7) or (8) is violated, we rescale our unconstrained estimates ($\eta$).

We minimize the amount of corrections required to find a solution for the BBCAv2 model through optimization. The amount of corrections is defined as the sum of the distances between the unconstrained ($\eta$) and corrected values counterparts (x) divided by the standard deviation of the unconstrained value ($\sigma$):

$$min. \sum\nolimits_{r \in \mathbf{R}} \frac{|x_r - \eta_r|}{\sigma_r} \qquad (9)$$

This nonlinear function is convex which guarantees a global minimum. We convert it to a linear problem, by introducing a slack variable z so that $\forall\, r \in \mathbf{R}$:

$$-z_r \leq x_r - orig_r \leq z_r \qquad (10)$$

Hence, our problem becomes:

$$min. \sum_{r \in \mathbf{R}} \frac{z_r}{\sigma_r} \qquad (11)$$

*st. Eqs* (6), (7), (8), (10)

The solution x is a vector with 3444 values and matches a total of 8613 linear constraints. An initial solution can also be found for each region, correcting the unconstrained estimates to match our constraints starting from leaf regions in the region hierarchy to the top-level brain regions (see S1 Document, S2 and S3 Figs). We use the simplex algorithm from the scipy python library [93].

We find that for 16% of the regions of the brain (shown in colors in Fig 6A and 6B), their $\eta$ estimates are incongruent with the rest of the brain. Their corrected values x are therefore not falling within the range $\eta \pm \sigma$. In some regions of the striatum (including caudoputamen), the number of GAD67 neurons from $\eta$ overshoots the estimated total number of neurons. Our optimization makes these regions fully inhibitory. This is in line with literature findings on the mouse striatum, which describe it as almost fully inhibitory [94]. Among the rest of the regions where significant corrections are needed, almost one-third are subregions of the hindbrain, for which we collected very few literature values. Conversely, few inconsistent first estimates are found in subregions of the isocortex, for which more literature data are available. The remaining inconsistencies for $\eta$ may have different sources:

- They can be explained by a poor estimate of the densities from the transfer function.

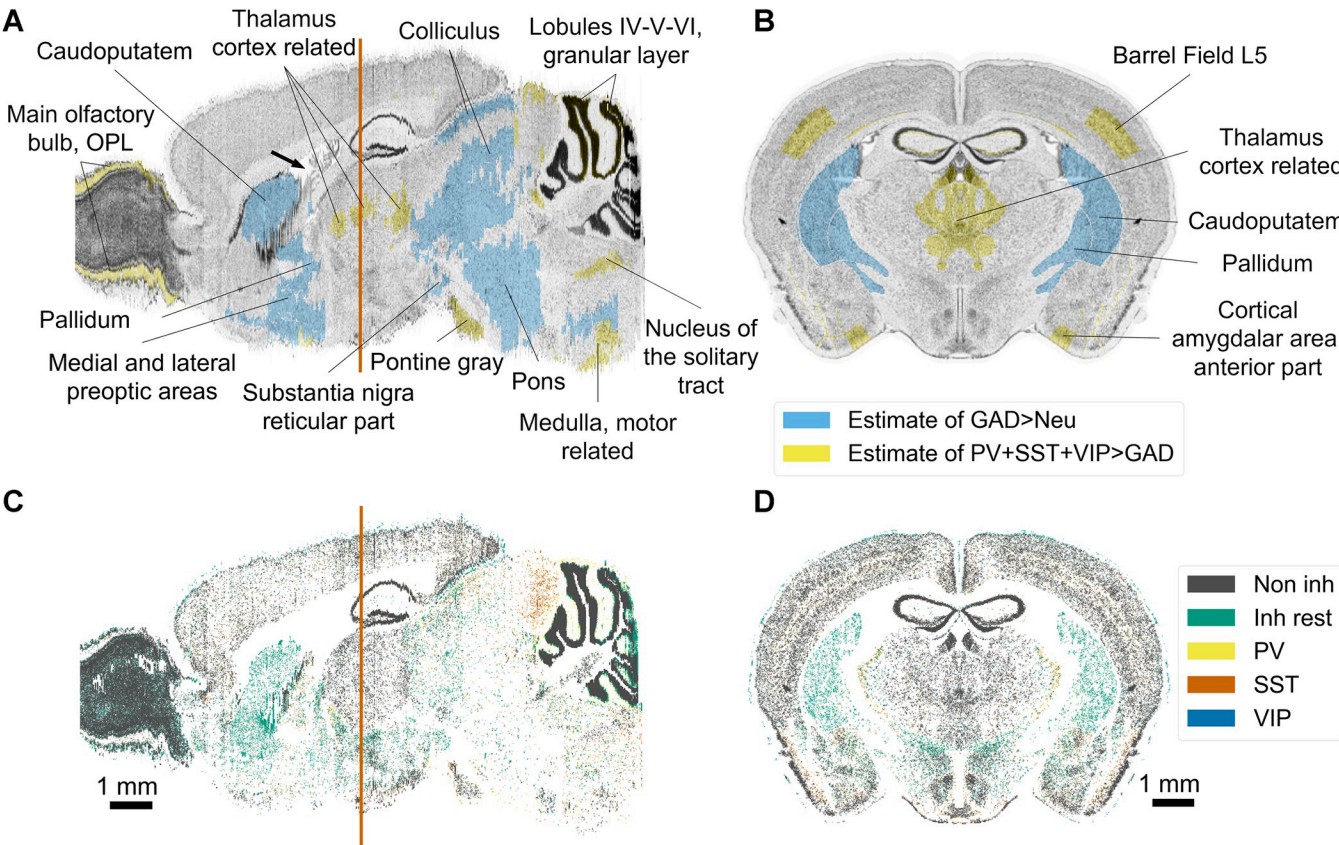

**Fig 6. Results of the density pipeline.** (A) Sagittal and (B) coronal view of the Nissl reference atlas used for BBCAv2, showing cells of the mouse brain. Regions with high cell density appear in dark grey. Regions that required significant corrections (see Section 2.7) for their neuron subtype density estimates are colorized (Neu = neuron). The coronal slice chosen is displayed as an orange line on the sagittal slices. Arrow in black shows the location of the lateral ventricle of the mouse brain. The blue line behind the sagittal slice highlights the drop of Nissl expression coming from the original Nissl experiment from Dong [18]. (C). Sagittal and (D). coronal view of the BBCAv2, showing the positions of the different types of neurons. PV+, SST+ and VIP+ cells appear respectively in yellow, dark orange and blue. The rest of the GABAergic population is color-coded in green and the remaining neurons in gray. The variation of the distribution of neurons is following the original distribution of Nissl expression.

- Divergent estimates from the literature for the same region might violate Eq (7). These regions would appear in yellow on Fig 6A and 6B).

- Our assumptions may not be accurate in some regions of the brain.

- The densities of neurons from the step 2 of the pipeline (see Fig 1), used in Eq (6) and (7), diverge locally from the literature findings.

After our corrections, we obtain consistent density estimates for PV+, SST+, VIP+, and GAD67+ neurons for each region of the AV.

## 2.8. Placement of the cells within the mouse brain volume

At step 4 of the pipeline (see Fig 1), we convert our density estimates into a list of cell positions for each region, using an acceptance-rejection algorithm. For each region, the algorithm samples voxels, according to the cell density distribution (computed at step 2), and assigns cells to them, until it reaches the target total number of cells in that region. The same algorithm can then be applied to label cells as neurons or glial cells as described in Erö et al. [17]. We further

subdivide neurons into PV+, SST+, VIP+, and GAD67+ neurons, following our density estimates. In brief, $\forall$ r $\in$ **R**, we select uniformly neurons within the voxels of region r, that we label as inhibitory, until the region contains the target number of nGAD_r inhibitory neurons. Similarly, we distribute the nPV_r neurons, in voxels of r containing GAD67 neurons, nSST_r in voxels of r containing non-PV inhibitory neurons, and nVIP_r in voxels of r containing the remaining inhibitory neurons. The non-inhibitory neurons are labeled as excitatory or purely modulatory neurons following Eq (1).

Finally, we assign to each cell a uniformly random 3D position within the boundaries of its voxel. The result of this procedure is shown in Fig 6C and 6D.

## 3. Results

### 3.1. An improved computational method to build an atlas of inhibitory neurons in the mouse brain

In this study, we present a substantially improved workflow to produce the second version of the Blue Brain Mouse Cell Atlas workflow (BBCAv2). The new workflow replaces the manual alignment of ISH datasets with the image alignment algorithm developed by Krepl et al. [56]. The pipeline also includes a new method to estimate cell densities from a large body of literature data. In total, we integrated values from 56 different literature sources.

All steps of this process are guided by a set of assumptions about the cellular composition of the mouse brain. Using our improved workflow, we are not only able to re-estimate the densities of inhibitory neurons in the mouse brain, but also to estimate the densities of their various subtypes. The result is an updated mouse cell atlas which provides density estimates and their standard deviations for each cell class in all regions of the CCFbbp AV. We additionally derived from our cell atlas densities a 3D position and a type label for each cell within the mouse brain (see Fig 6C and 6D). The same methods can also be applied to other cell types to deepen our knowledge on cellular composition of the mouse brain. The new version of the Blue Brain Mouse Cell Atlas is available on the Blue Brain Portal website (https://bbp.epfl.ch/nexus/cell-atlas). The code used to produce these results can also be downloaded from https://github.com/BlueBrain/atlas-densities.

### 3.2. Distribution of inhibitory neuron subtypes for the whole mouse brain

The new cell atlas has updated estimates for the densities and numbers of inhibitory neurons, based on more literature data. In addition, the new atlas differentiates PV+, SST+ and VIP + neurons and provides their densities and absolute numbers for each region of the mouse brain. An overview of the results is shown in Fig 7 and Table 3. The complete list of densities for all cell types in all regions of the Allen Mouse Brain Atlas is available as supporting information (see S2. Excel). We estimate that the mouse brain holds approximately 14.55 million of inhibitory neurons in total (see Table 3). GABAergic neurons make up 20.27% of the total neuron population, which is significantly larger than the previous estimate of 15.68%, using the BBCAv1 workflow. We find that 3.57% of neurons are PV positive, 3.20% are SST positive, and 0.63% are VIP positive (see Fig 7A and S2 Excel). This leaves a residual GABAergic neuron population of 12.87%, which represents almost two-thirds of all inhibitory neurons.

The ratios of our inhibitory neuron subtypes counts divided by the neuron counts of the BBCAv2 fall near the numbers reported by Kim et al.: 4.03% PV+, 3.29% SST+, 0.63% VIP + neurons [7]. The differences can be explained by the integration of other sources of literature to estimate these neuron population densities in the brain.

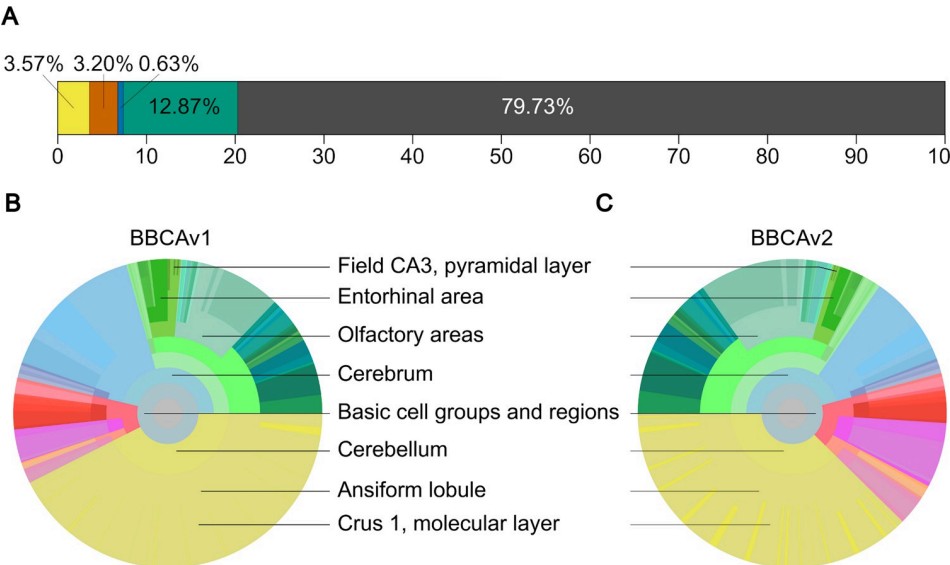

**Fig 7. Distribution of the GABAergic neurons in the mouse brain.** (A) Ratios of PV+, SST+, VIP+, GAD67+, and remaining neurons of the mouse brain. InhR cells appear in green, PV+ in yellow, SST+ in orange, and VIP+ in blue. The remaining neurons appear in grey. (B), (C) Circular distribution of GAD67+ neurons in different regions of the mouse brain according to different parcellation schemes ranging from coarse at the center to fine-grained at the periphery (colors and disposition similar to Fig 4A). (B) displays results of BBCAv1 and (C) results of BBCAv2.

Compared to BBCAv1 (Fig 7B), the number of inhibitory neurons in BBCAv2 (Fig 7C) is larger in almost all regions of the brain (except notably, the hippocampal formation and the striatum). The estimated number of inhibitory neurons in the cerebellar cortex granular layer increases from 203,000 in the previous pipeline to 867,000. Similarly in the isocortex, this number rises roughly by 53% (see S2 Excel). Finally, the BBCAv2 estimates 2.05 million inhibitory neurons in the olfactory areas.

**Table 3. Estimates of inhibitory neurons in the mouse brain in millions and percentage (%) subtypes.**

| Brain Region | BBCAv1 | BBCAv2 | | | | | |
|---|---|---|---|---|---|---|---|
| | Inh. | Inh. | $\frac{Inh}{Neu}$ (%) | $\frac{PV}{Inh}$ (%) | $\frac{SST}{Inh}$ (%) | $\frac{VIP}{Inh}$ (%) | $\frac{Rest}{Inh}$ (%) |
| Isocortex | 1.44 | 2.20 | 24.7 | 23.2 | 25.0 | 12.6 | 39.2 |
| Olfactory areas | 1.45 | 2.05 | 25.1 | 1.3 | 13.8 | 4.7 | 80.2 |
| Hippocampal form | 0.51 | 0.49 | 9.9 | 12.6 | 33.9 | 9.0 | 44.5 |
| Cortical subplate | 0.12 | 0.22 | 43.8 | 2.6 | 28.3 | 3.7 | 65.4 |
| Striatum | 1.72 | 1.40 | 81.4 | 1.2 | 17.5 | 0.2 | 81.1 |
| Pallidum | 0.18 | 0.23 | 91.8 | 6.9 | 28.7 | 0.2 | 64.2 |
| Thalamus | 0.19 | 0.26 | 18.9 | 15.2 | 32.7 | 0.0 | 52.1 |
| Hypothalamus | 0.42 | 0.50 | 41.4 | 3.0 | 29.0 | 0.4 | 67.6 |
| Midbrain | 0.39 | 0.99 | 76.4 | 12.6 | 37.4 | 0.9 | 49.1 |
| Hindbrain | 0.26 | 0.66 | 62.5 | 23.9 | 38.2 | 0.6 | 37.3 |
| Cerebellum | 4.79 | 5.47 | 13.0 | 28.8 | 0.8 | 0.1 | 70.3 |
| Whole Brain | 11.25 | 14.55 | 20.3 | 17.6 | 15.8 | 3.1 | 63.5 |

Inhibitory neuron counts (in millions) extracted from the cell atlas generated using the BBCAv1 and BBCAv2 pipelines. The second column of the BBCAv2 corresponds to the ratio of inhibitory neurons according to the total number of neurons in this region. The following columns display the proportion of each inhibitory neuron subtype according to the inhibitory neuron population.

### 3.3. New Excitatory/Inhibitory ratios for the isocortex

The BBCAv2 also provides new estimates for the ratio of inhibitory/excitatory neurons (see Fig 8). For example, we predict that 25% of the neurons of the mouse's isocortex are inhibitory (see Table 3 and Fig 8B). Thus, we estimate a higher proportion of inhibitory neurons than previously reported. For rodents, literature estimates are around 20% inhibitory neurons [14,95].

The poor alignment of the AV to the Nissl dataset counterpart (including the layer 1 layer 2 boundary, see Section 2.2) may explain this result. It is artificially raising the density of neurons in layer 1, which in turn, increases the ratio of inhibitory neurons in the isocortex (black contour in Fig 8B). Additionally, the densities of PV+, SST+ and VIP+ neurons, mostly constrained by Kim et al. [7], are providing a lower limit for the total number of inhibitory neurons in each region (green bars in Fig 8B). To improve on the ratios of inhibitory neurons in the isocortex we should therefore check its densities of neurons and our predictions of GAD67 neurons in those regions.

We also notice that the estimated densities of inhibitory neurons are quite similar from layers 2 to 5, and across isocortex subregions (Figs 8A and S6). As an additional validation, we check if our model can reproduce the findings of Meyer et al. on rats in the barrel cortex [95]. To do so, we computed the neuron density profiles according to the distance between their soma position and the pia (see additional methods and Fig 8C). The direct comparison of our density profile and the results from Meyer et al. are shown in Fig 8D. The distributions of the excitatory and inhibitory neuron populations according to their distance to the pia are similar despite the cross-species comparison.

### 3.4. Validation of the pipeline

In our revised workflow for BBCAv2, we used a linear transfer function to map ISH expression levels to literature cell density to then estimate the cell densities in regions where no literature value is available. The fidelity of the transfer function depends on both the quality of the ISH images and the quality of the underlying literature estimates of the cell densities. As we highlighted in Section 2.3, the genetic marker datasets are prone to artifacts due to slicing. Also, their coverage of the mouse brain is not perfect (see Fig 3D). Combining multiple datasets of the same genetic marker can blur the spatial details of brain structures. For each marker, the AIBS website hosts similar ISH experiments in sagittal (S7 Fig) and coronal (Fig 3A) slices. We compare the results of our pipeline, using two different coronal ISH datasets for GAD67 and find very similar results in terms of fitting, distribution of inhibitory neurons and remaining errors after corrections (see S2 Excel). To blur the individual artifacts of the ISH datasets, we combine these datasets after realignment, taking the mean of their expressions. We then linearly fit the marker expression from the resulting dataset to literature densities, but we find no significant improvement. However, changing the method to extract ISH data will only impact the x axis of Fig 5. The spreading of points along the y-axis will not be resolved (e.g., when multiple literature sources reporting density values in the same region contradict each other, see Fig 4B).This result can be explained by the slicing of these two experiments which, in both cases, is not fully covering the tips of the olfactory bulbs and the cerebellum as illustrated in Fig 3D.

Next, we explore the impact of the amount of literature values available on the quality of the transfer function.

We do this experimentally by running our workflow with a fraction f of the available literature points (70% up to 95%). For each fraction f, we perform 20 separate experiments, each with a different random subset of literature values. Thus, for each fraction f, we obtain 20

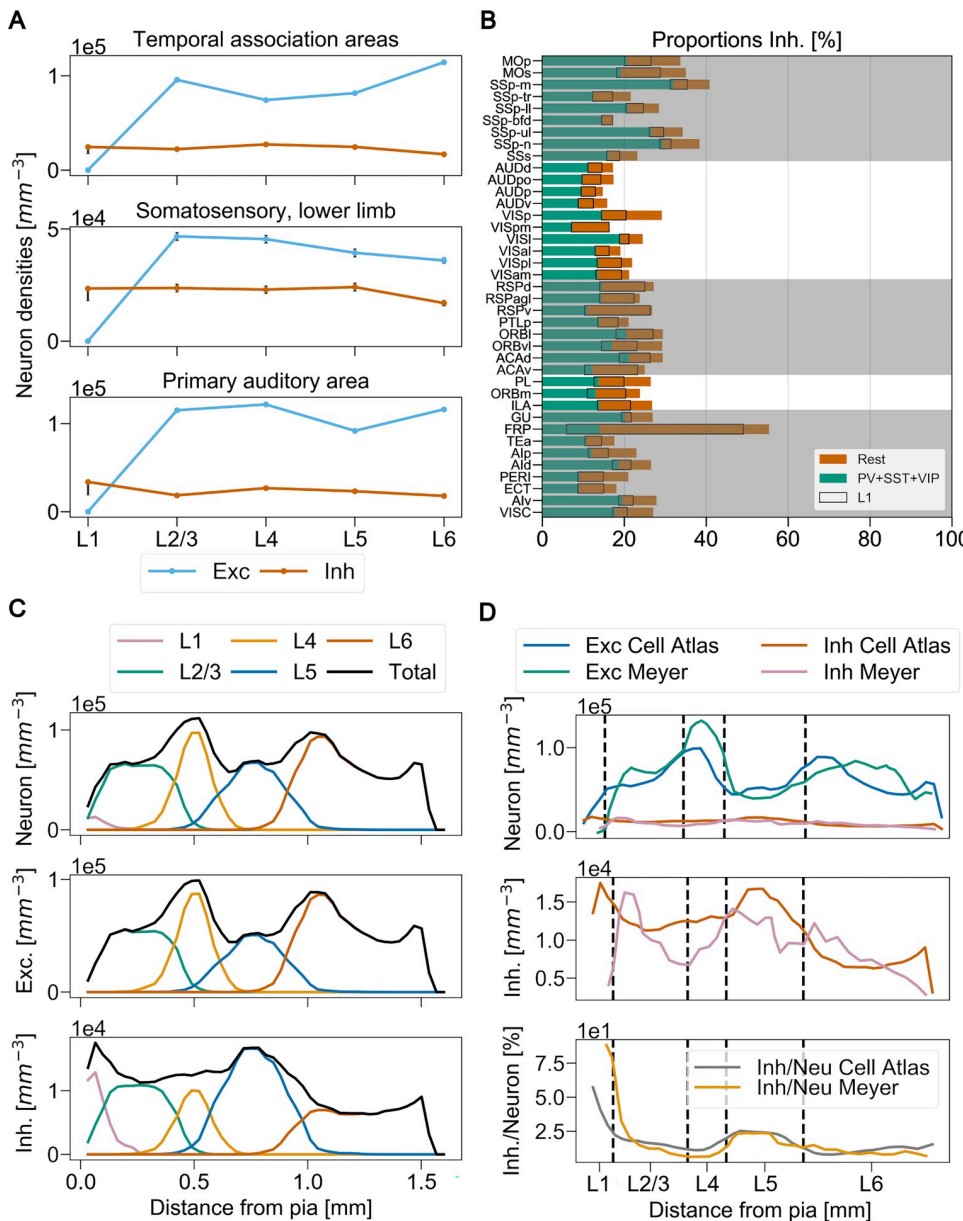

**Fig 8. Cortical excitatory/inhibitory ratios.** (A) Relative cortical layer cell densities from the primary somatosensory cortex lower limb (SSp-ll), primary auditory cortex (AUDp) and temporal association areas (TEa). Blue lines correspond to excitatory neuron densities (Exc) and the orange lines inhibitory neuron densities (Inh) in cells/mm³. Final confidence intervals are also displayed for each layer as vertical lines. (B) Ratios of inhibitory/excitatory neurons (Inh/Exc) across anatomical regions of the isocortex, expressed as percentage and arranged in five subnetworks based on Kim et al. [7]. Acronyms correspond to the AV naming convention [60] (see Table A in S1 Document). The portion of inhibitory neurons that belongs to L1 is shown with a black contour. (C) Somatosensory barrel field neuron distributions according to pia. Each layer is represented by a different color and the total density according to distance to pia appears in black. (D) Somatosensory barrel field excitatory/Inhibitory neuron distributions (Exc/Inh) according to pia compared to the results extracted from figures 2EFG from Meyer et al. [95]. AV mean layer limits are represented as dash lines and aligned to the ones provided by Meyer et al.

different estimates of the cell density in each brain region r ∈ **R**. We also compute the standard deviation of the resulting density values for each region ($\sigma_{(r, f)}$). As we increase the number of literature values that are contributing, we expect the standard deviation of the density

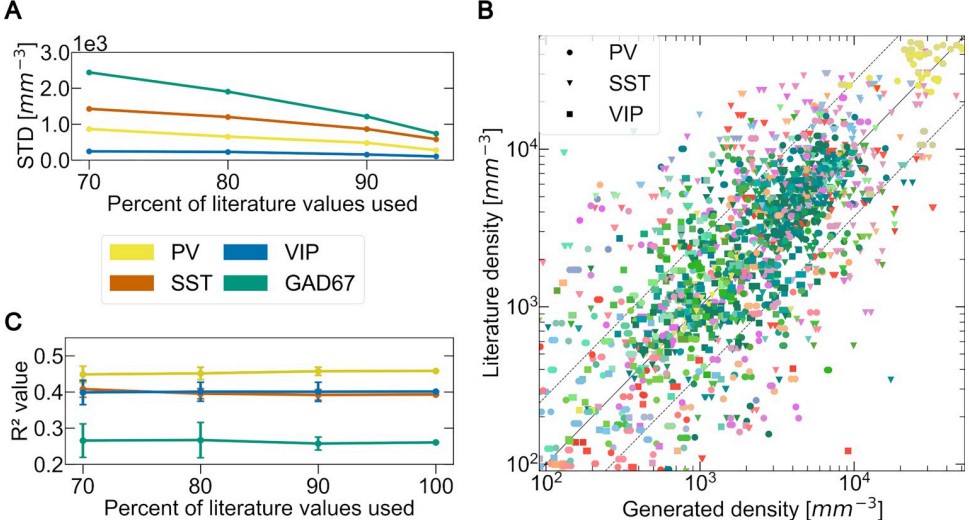

**Fig 9. Impact of the amount of literature on the BBCAv2.** (A) Evolution of the mean standard deviation of the BBCAv2 generated densities for each genetic marker (shown in different colors) using different percentages of the total amount of literature values available. (B) Comparison of the generated density values of the BBCAv2 with their literature counterparts using 90% of the total amount of literature values available (20 trials). The color encodes the brain regions according to the AV, while the shapes of the points encode for cell types. The middle line delimits equal quantities, while the dashed line shows the average deviation of 2.7-fold between literature values reporting on the same region for PV, SST and VIP neurons. (C) Evolution of the mean coefficient of determination $R^2$ of the fitting, for each genetic marker (shown in different colors), performed using different percentages of the total amount of literature values available.

estimates to decrease. We observe that the average of all standard deviations for each marker gets smaller as more literature values are integrated (see Fig 9A). This finding supports the existence of a ground truth for the cell densities in the mouse brain that our algorithm is targeting.

For each region r, we then use all $\sigma_{(r, f)}$ to extrapolate the standard deviation of the density for f = 100%. This standard deviation $\sigma_{(r, 100\%)}$ represents the confidence of our pipeline in our final predicted densities for each region (see S2 Excel).

Surprisingly, the quality of the fitting does not significantly improve with more literature values added to the pipeline (see Fig 9C). This result indicates that the coefficient of determination is not mostly influenced by the literature values quantity. R2 could however, be dependent on the neuron distribution obtained at step 2 (see Fig 1), the ISH experiments or the method chosen to group the regions (here grouping by main regions: cerebellum, isocortex and rest, see Fig 5).

We also test the capacity of the BBCAv2 to predict values of literature. We select 90% of the literature points available, and then compare the generated densities to the remaining points (see Fig 9B). Most of the density values produced with this method fall within the confidence interval of their literature counterparts. This workflow is therefore capable of predicting densities of cells within range of what literature provides.

As an additional qualitative validation, we compare our generated GABAergic neurons positions with a sagittal slice of GAD67 expression from the AIBS (see Fig 10A and 10B) and find a similar whole brain distribution. Some of the visual differences between Fig 10A and 10B can be explained by the fact that in the image from the BBCAv2 (Fig 10B), the soma sizes are uniform across all cells, while in the brain, soma sizes differ considerably between cell types and brain regions (Fig 10A). Moreover, the AV1a is based on the coronal Nissl

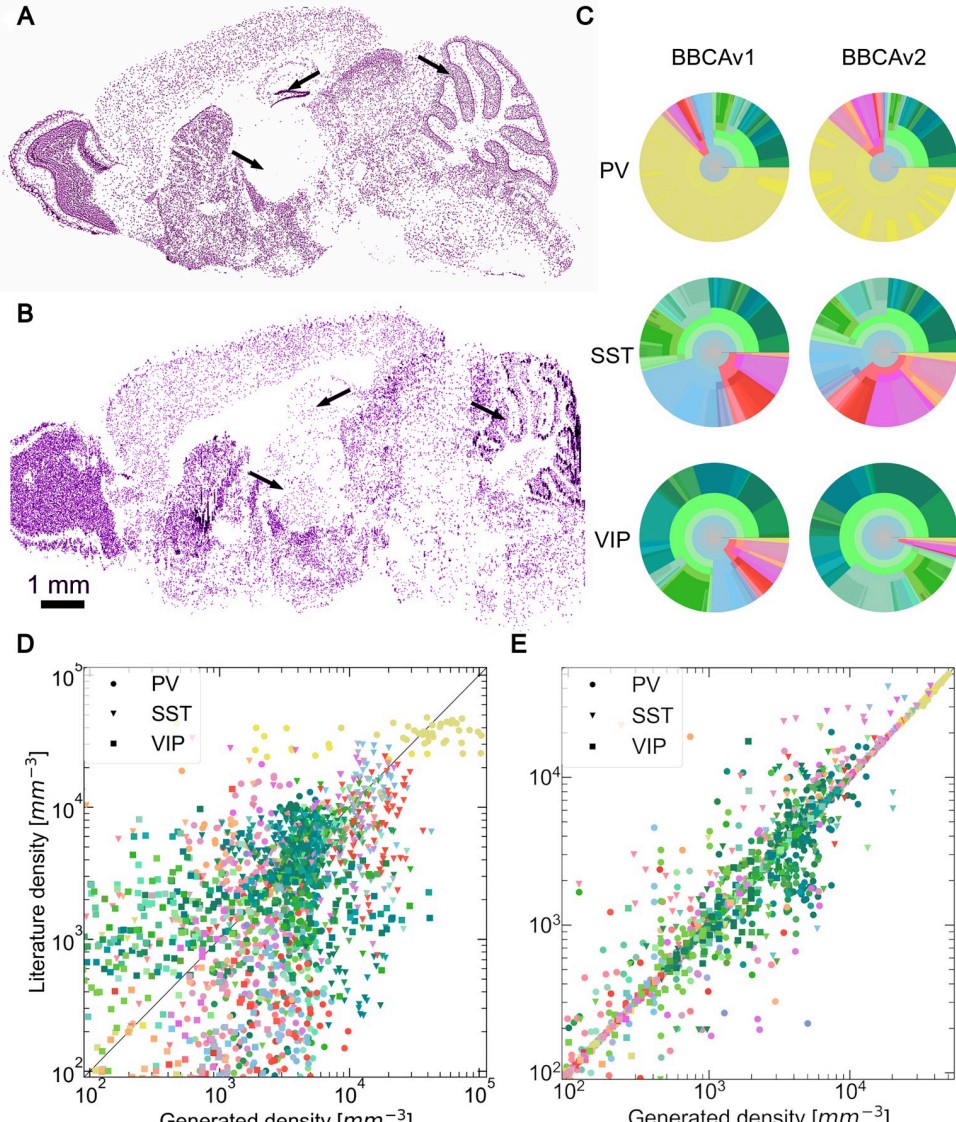

**Fig 10. Qualitative validation of the BBCAv2 and comparison with BBCAv1.** GAD67 sagittal brain slices comparison between (A) the filtered image from an ISH experiment by the AIBS (experiment No 480—slice 13) at the top and (B) the BBCAv2 predicted GABAergic neurons positions at the bottom. Cells reacting to the marker are displayed in purple. Blank regions in our model correspond to fiber tracts regions which host no neuron. This slice does not take regional soma sizes into account. Most of the density features of the real experiment can be found in the BBCAv2 slices. Black arrows point to the thalamus (left arrow), molecular layer of the dentate gyrus (center arrow) and granular layer of the cerebellum (right arrow) of each brain. (C) Circular distributions of the densities of PV+, SST + and VIP+ neurons computed according to the Erö et al. method on the left [17] and our method on the right (colors and disposition as in Fig 4A). Distribution of PV+, SST+ and VIP+ neuron density values reported in literature against generated densities, using the BBCAv1 method (D) and the BBCAv2 method (E), for similar regions. The color encodes the brain regions according to the AV, while the shapes of the points encode for cell types. The middle line delimits equal quantities. In (E) the model is built on the regional literature references used in the figure which explain the numerous points on the middle line.

experiment from the AIBS (see Section 2.2) which is not well covering the tips of the cerebellum and the olfactory bulbs (see Fig 6A) and deforming them. However, this comparison highlights the fact that in our model the thalamus and the cerebellum have too many inhibitory neurons. Additionally, the granule layer of the dentate gyrus is clearly visible on the original

experiment but not in the BBCAv2 brain slice (center black arrow). Our density estimates for this region (5403 mm$^{-3}$]) are however to be in the range of the literature [48,77]. The granule layer of the dentate gyrus is very dense [17], and Zeisel et al. [69] have shown that its excitatory neuron population expresses GAD67 at a low level, but not GAD65. A closer inspection of the GAD67 and GAD65 ISH brain slice images (see S7 Fig) allowed us to identify two types of cells expressing the markers. The cells with the largest radius and the strongest level of expression of GAD67 and GAD65 are sparse and their description corresponds to the inhibitory neurons, as in Jinno et al. study [77]. The other population of cells is smaller in diameter and their expression of GAD67 is lower compared to the other population. This population is additionally not visible in the GAD65 brain slice, which indicates that it corresponds to the excitatory neuron population.

In Fig 10, the original slice in panel A shows spatial detail with higher resolution than our reconstruction in panel B. For example, in the olfactory bulb, panel A shows the layering of cells, as we do in the cerebellum and hippocampus. The spatial resolution of our cell atlas is lower, because the density is assumed uniform within regions at the lowest level of the AIBS region hierarchy (assumption 5), following the estimated cell density.

We also compare the estimates of PV, SST and VIP densities from the BBCAv2 against similar estimates with the BBCAv1 [17]. For the BBCAv1, we use the global number of PV, SST and VIP neurons in the mouse brain from Kim et al. and the PV, SST, and VIP realigned datasets (see Section 2.3, Fig 10C and S2 Excel) [7]. The BBCAv1 maintains the global number of inhibitory subtypes expected in the whole brain at the cost of losing the local distribution (see Fig 10D). For instance, Kim et al. [7] found that the cerebral nuclei regions contain 62 VIP neurons per mm$^3$, while the BBCAv1 predicts it to contain 963 VIP neurons per mm$^3$ (see Fig 10C, cerebral nuclei regions appear in light blue). The BBCAv2 method, on the other hand, aims to maintain local estimates from literature in a coherent atlas (see Fig 10E). It is therefore usually providing density estimates closer to their literature counterparts for each neuron type.

Our model integrates literature values in the coherent framework of the cell atlas, with constraints and assumptions that we control and that we can refine whenever more data become available about the mouse brain.

## 4. Discussion

In this study, we show that our new method is capable of estimating inhibitory neuron counts for each region of the mouse brain, including PV+, SST+, VIP+ neurons, and remaining inhibitory neuron subtypes, in a consistent framework: The Blue Brain Mouse Cell Atlas (BBCAv2). The results of our method are constrained by literature findings (see Section 2.5), ISH experiment data from the AIBS (see Section 2.3) and simple assumptions (see Section 2.4). BBCAv2 extends the results of BBCAv1 and deepens our knowledge of the cellular composition of the mouse brain. Additionally, the BBCAv2 can be easily adapted to estimate the densities of other cell types, providing more details about the composition of the brain in terms of morphological, molecular, or electrical properties.

### 4.1. A new approach to integrate literature findings into the Blue Brain Cell Atlas

We presented a new approach to reach a consensus on cellular composition in the mouse brain based on local literature findings instead of global constraints. While most whole-brain studies evaluate the densities of a single cell type [7–10], our method integrates multiple cell types and creates a framework where these densities are coherent with respect to each other. To compensate for the great variability of cell density measurements in literature and to avoid

a bias towards any particular finding, we incorporated as many data sources as possible (see Section 2.5). The BBCAv2 therefore reflects our current knowledge about cell composition in the mouse brain. However, this makes it also more difficult to validate, since only limited data or studies remain available to which to compare our results. Moreover, any additional literature value that we could use to validate our counts might itself only be as trustworthy as the literature values used to build our model. On the other hand, qualitative validations can highlight inconsistencies in our results as shown in Fig 10A and 10B.

## 4.2. Reference dataset's impact on cell densities

Both Blue Brain Mouse Cell Atlas pipelines are based on a pair of reference atlas datasets from the AIBS which exist in multiple versions. We assessed newer versions of the AV and analyzed their impact on cell densities calculation.

**4.2.1. Alignment of the pairs of reference volumes.** We found that the best cell density estimates can be obtained if the pair of reference datasets (Nissl volume and AV) is perfectly aligned to each other. For example, if the border between two regions with significantly differing densities (e.g., isocortex L1 and L2, see Fig 2A and 2C) is not perfectly aligned with the corresponding Nissl, then the densities in the particular regions will be over- or underestimated. Hence, the CCFv3 AV, while depicting a smooth average brain volume, provides the least reliable density results when combined with Nissl2.

**4.2.2. Smoothness of the annotation volume.** The Nissl reference volumes are stretched and sometimes torn by the slicing, which yields a very noisy dataset. The derived annotation atlas had to follow this roughness with sometimes discontinuous region borders. The resulting AVs are therefore less attractive for circuit modeling. We can still benefit from the best of each dataset, computing the densities using the CCFv2 or CCFbbp and then populate the CCFv3 smooth brain volume with these estimates. As for our assumption 5, cell distribution in the lowest level of the region hierarchy will be considered homogenous. Potential inner region gradient can be kept if a common vector base is defined in each AV version. Another solution could involve the realignment of the AV2 to the AV3. With this work, we will recover the sub-regions of the AV2 that were not found in AV3 (e.g., cerebellar cortex, hippocampus CAs).

**4.2.3. Remaining artifacts in the Nissl volume.** In Fig 6A, we can also see another flaw of the Nissl dataset in the form of a drop of expression in slices between the olfactory bulbs and the thalamus which seems to be unrelated to the regions they are crossing (see also S4A–S4C Fig for more details). This drop of expression is then clearly visible in the resulting distribution of neurons in Fig 6C. Yet, the neuron count estimates are used in the rest of the pipeline as a top constraint for our inhibitory neuron distribution, which can partially explain some of the errors we encounter at the final stage. Finding other Nissl datasets to create an average Nissl volume or exploring image processing techniques such as histogram matching and blurring should be part of future works to improve the model.

## 4.3. Challenges linked to ISH marker expression datasets

We applied Krepl's algorithm [56] to realign the filtered ISH images of the mouse brain from the AIBS at step 1 of our method (see S8 Fig). The resulting datasets are crucial for the BBCAv2 pipeline as they allow cell density estimation in the regions missing literature values.

**4.3.1. Assumptions about the ISH datasets.** First, we considered the ISH image slices to be perfectly vertical to simplify their realignment to Nissl and the interpolation between them. This assumption is however not true and the AIBS provides the position of the image corners in each ISH experiment metadata. In a future iteration of our pipeline, we will consider

integrating each ISH slice at its exact position as it might help the algorithm to find the right landmarks to realign the images to the Nissl counterpart.

We also chose to use an all-or-none filter on the filtered images because we were interested in counting all the cells expressing the marker. However, we could deduce counts of subpopulations of neurons which express their marker at specific levels. These levels might depend on the regions or the cell types, but it is an opportunity to improve the BBCAv2 locally.

**4.3.2. Brain coverage of the ISH datasets.**   The ISH experiments partially cover the whole mouse brain, with one slice stained per 200 μm in the rostro-caudal axis (see Fig 3D) [19]. Additionally, they represent different mouse individuals whose neuron distribution might vary significantly with respect to one another. As we discussed in Section 3.4, the combination of similar ISH datasets can improve their coverage of the brain and blur out the remaining artifacts. Furthermore, ISH experiments performed with a sagittal slicing covers regions not included in the coronal experiments and their addition will improve the ISH whole brain coverage. Using these datasets requires the realignment of the sagittal images of the inhibitory markers to the Nissl volume, using the deep learning algorithm described from Krepl et al. [56]. The usage of these datasets is currently investigated for a future update of the pipeline.

**4.3.3. Remaining artifacts of the ISH datasets.**   The marker datasets, despite having been processed by the AIBS, are still subject to artifacts and noise which will artificially raise or lower our cell count estimates (see Fig 3A). To counter these effects, Erö et al. [17] used the Nrn1 marker, as it is expressed in excitatory neurons, to limit the expression of GAD67. The authors normalized the expression of GAD67 with the sum of the expression of GAD67 and Nrn1 and multiplied the result with the ratio of inhibitory neurons using global counts from Kim et al. [7]. The GAD65 marker has also been reported to be expressed in GABAergic cells [96]. Similarly, we can extend our approach, knowing the density of neurons expressing Nrn1 and GAD65 in regions of interest, to improve our estimates of GABAergic neurons.

**4.3.4. Alternative methods to estimate cell counts from ISH images.**   Future analysis will include estimating cell density from the ISH images using automatic point-detection algorithms (out of the scope of the present study) as presented in Erö et al. [17]. These algorithms will provide new cell counts to rectify our estimates. However, we will have to solve the issue of cell overlapping, especially in high density regions such as the cerebellum, as discussed in Erö et al. [17]. Moreover, cell counting requires the realignment of the high-resolution images from the AIBS to the Nissl and annotation of the 25 μm resolution volumes in order to delimit the regions of the brain in the image slice. The difference in resolution makes the realignment more difficult and the borders of the regions would be less precise due to the low resolution of the associated reference volumes. In addition, the cell counting methods are more sensitive to noise and artifacts that are still present in the ISH high resolution images.

## 4.4. Limitations of the assumptions of the pipeline

The BBCAv2 estimates of the different cell densities are based on the five assumptions described in Section 2.4. These assumptions come from findings mostly published on the isocortex and represent the rules we used to categorize all cells in the mouse brain [55,58,69–74]. However, for each of our first four assumptions, we found exceptions reported in literature:

1. A small portion of astrocytes from the visual cortex have been reported to express GAD67 [49].

2. Szabolcsi and Celio [97] observed that ependymal glial cells express PV. These cells are surrounding the ventricles of the mouse brain whose regions are present in the AV. No neurons were placed within these regions as can be seen in Fig 6A and 6C; these cells should therefore have a minor impact on our results.

3. Zeisel et al. [69] reported that PV, SST, and VIP populations overlap in most regions of the brainstem. There are many types of cells co-expressing at least two of our markers with different levels of expression and we found no automatic solution to detect each of these cells using the data available, since the ISH data from the AIBS comes from different species. Lee et al. [98] also pointed out, using double ISH staining, that a population of neurons in the isocortex is co-expressing PV and SST.

4. Zeisel [69] also found that PV and SST are expressed in excitatory neurons and VIP in modulatory and non-inhibitory neurons in some regions of the brainstem such as the colliculus and medulla, which were detected as errors in our reports (see Fig 6A).

If validated by additional studies of density estimates in the regions where these exceptions have been reported, we could justifiably subtract a proportional amount from the deduced numbers of GABAergic cells in future versions of the model.

As we describe in Section 4.2, some of the inner region variation of cell densities can be lost applying the assumption 5. This concerns gradients within the neuronal population since the neuron distribution is derived from the Nissl volume. For instance, in the hippocampus CAs, Jinno et al. [77] have shown a dorsal to ventral distinction in the density of the inhibitory neurons, which is not reflected by the subdivision of the AV. Additional subregions for the AV have to be created to include these findings.

## 4.5. Reliability of quantitative studies on inhibitory neurons

One of our objectives was to match regional estimates from literature with our neuron types' density distribution. To do so, we performed a literature review of quantitative studies on PV +, SST+, VIP+, and GAD67+ neurons in the mouse brain (see Section 2.5).

**4.5.1. Literature coverage of the mouse brain on inhibitory neuron density.** We found that literature does not provide a complete coverage of the whole mouse brain for GAD67. This can be explained by the difficulty performing cell counting or cell detection on large tissue volumes, or by the fact that some regions are more frequently studied than others (e.g., barrel cortex, hippocampus CAs, see Fig 4B). Moreover, GAD67, PV, SST and VIP are extensively used genetic markers in literature, which is not the case for most of the markers available (e.g., LAMP5). This means that the more precise our cell composition becomes the more difficult it will be to gather enough data points from literature and obtain meaningful transfer functions (see section 2.6).

**4.5.2. Variability of the literature estimates on cellular composition.** We also found that studies reporting similar measurements of neuron densities sometimes yielded very disparate results (see Fig 4B and Table 1). Literature density estimates can be biased by the age, sex, staining technique, neuron counting method, or measurement of the volume of the region of interest [7,11].

Some of the densities used to build our transfer functions were computed according to cell counts from literature divided by the volumes of the regions of the AV, and potentially prone to errors as we pointed out in Section 2.2. Some regions are also known to be purely inhibitory (e.g., layer 1 of the isocortex) and we therefore used the Cell Atlas neuron densities, which are themselves subject to errors as discussed previously. Future work on the reference atlases should improve the reliability of the literature estimates.

## 4.6. Variability of the transfer functions

Fitting a transfer function from region mean intensity to cell type density is a required process to obtain the density of inhibitory cell types in regions where no literature data are available (see Section 2.6).

**4.6.1. Factors impacting the quality of the fitting.** We found that the fitting of the transfer functions from marker expression to cell densities is sensitive to errors linked to the datasets used as input: literature variability (see S9A Fig), sparseness (see Fig 3D) and misalignment of the ISH data according to the AV (since these are realigned to the Nissl volume). The variability observed in Figs 5 and S9 might also be linked to the different cell sizes which impacts the region mean intensity. All of these effects can explain some of the discrepancy in BBCAv2 results and the fitting's low coefficient of determination.

**4.6.2. Alternative methods to fit region intensity to neuron density.** In future work on the BBCA, the transfer functions can be improved by refining the groups for which separate fittings will be done. These subgroups could correspond to different expected soma sizes for the cell type of interest. However, this would reduce the number of points available to fit the function and might result in an overfit.

Other monotonically increasing functions can also be tested to fit the cloud of points of Fig 5 such as sigmoid functions to integrate any threshold dynamics for low expression values or saturation for high expression values.

## 4.7. Consolidation of neuron density estimates

As we discussed in Section 4.5, literature reliability on estimates of cell counts is currently insufficient. Similarly, the fitting of these literature values to marker expression suffers also from the lack of coverage and remaining artifacts of the ISH datasets (see Section 4.3). We described a method in Section 2.7 to reconcile disparate density estimates into the consistent framework of the BBCAv2. This method aims to minimize the amount of correction needed to obtain a coherent solution.

**4.7.1. Minimization function to limit corrections applied to estimates.** Each first estimate needed to be optimized to fit into a coherent model. To choose which values required the most corrections, we used our minimization function (11) to give a weight to each value. This weight is inversely proportional to the standard deviation of our first estimates ($\sigma$). Hence, this parameter accomplishes two purposes: first, it reduces the impact of dense regions on the global minimization score, and second, it gives more weight to estimates where either literature or the fitting are more confident. In this way, each weighted estimate participates in the minimization function according to its reliability. Therefore, we also minimize the bias towards any literature or fitting source, or any method to extract the first estimates in general with our optimization algorithm.

Other $\sigma$ values can be tested in the future, such as the total number of neurons in the region to limit the impact of neuron density and in turn, give the same relative weight to each region. Similarly, we can measure the impact of adding an extra weight to maintain the proportion of each cell type within a region. This new constraint will have to be balanced with the previous one.

**4.7.2. Bias of the BBCAv1 towards the BBCAv2.** We assumed that the distribution of neurons in the mouse brain using the BBCAv1 method is correct, which means that whenever there was a contradiction between this distribution and the BBCAv2 inhibitory neuron estimates (coming from literature or the fitting), the error will be attributed to the latter. BBCAv2 results are therefore influenced by BBCAv1. The results of Erö et al. have been validated against literature counterparts [17] but might be incorrect in some specific regions (as depicted in Fig 10C). The BBCAv2 seems however to be more in agreement with literature than the BBCAv1 (see Fig 10D and 10E). In future work, we will therefore test our new approach to estimate cell, neuron and glia densities in the mouse brain (step 2 of the pipeline—see Fig 1) based on literature and the fitting. To this end, we will extend the literature review from Keller et al.

[12]. The optimization process from step 4 will have to include these new constraints and variables to solve them all at once.

**4.7.3. Bias from the Kim et al. study on the BBCAv2 results.** The biggest contributor to our list of density values extracted from the literature (62% of all literature values—856 out of 1376) is the Kim et al. study [7] which provides density estimates for PV+, SST+ and VIP+ neurons. Moreover, these density estimates cover the entire brain and are consistent with each other (same species and the same methodology were chosen to count cells), which simplifies their integration. Our results are therefore correlated to the Kim et al. study (see Section 3.2 and Fig 10E). Any bias present in the Kim et al. study would impact largely our estimates for the three inhibitory subtypes. In contrast, our collected GAD67+ density estimates from the literature comes from various independent studies which can explain the more important variability of our resulting inhibitory neuron estimates (see Fig 9A).

We have tried to minimize the bias towards any literature source (and the Kim et al. study in particular) with our optimization algorithm which re-weights all literature estimates according to their reliability (see Section 4.7.1). Clearly, the results from our pipeline will become more reliable when brain-wide measurements of GAD67+ cell distributions become available.

## 4.8. Changes in counts of inhibitory neuron in the brain

We obtained a new distribution of inhibitory neurons in the mouse brain, including a new distribution of PV+, SST+, and VIP+ neurons, and the remaining inhibitory neuron population (Rest).

**4.8.1. Increase in the proportion of inhibitory neurons from the BBCAv1 to the BBCAv2.** Our new method also accounts for the Rest population. Inhibitory neuron counts in BBCAv2 are therefore higher compared to BBCAv1. The three regions with the most pronounced updates of inhibitory neuron densities are: isocortex, olfactory areas, and cerebellum (see Table 2 and S2 Excel). In the cerebellum, only the granular layer was updated since the molecular layer was already marked as purely inhibitory in both versions of the BBCA. As we observe in Fig 10A and 10B, this BBCAv2 model seems to display too many inhibitory neurons in the cerebellar cortex, granular layer and in the thalamus in general when compared to the GAD67 ISH experiment. Some interneurons of these regions might co-express PV and SST as pointed out by Zeisel et al. [69], and as can been seen with a manual inspection of ISH slices of PV, SST and GAD67. As we discussed in Section 4.4, we will implement this specificity in our assumptions. In the olfactory areas, we collected density estimates from literature [35,45]. These estimates compensate for the poor coverage of the olfactory bulb by the GAD67 marker experiment (see Fig 3D) and result in larger inhibitory cell density estimates in BBCAv2. In the isocortex, the BBCAv2 estimates an overall higher proportion of inhibitory neurons. In some regions, such as the barrel field or the auditory areas (see Fig 8B), the ratio of inhibitory neurons in the isocortex seems to be close to the 20% reported in Markram et al. [14] and Meyer et al. [95]. In other regions, such as the motor areas, the ratio of inhibitory neurons averages above 30%. For most of the isocortex regions, we gathered at least one density estimate for PV, SST and VIP expressing neurons from the literature; the ratio of inhibitory neurons in these regions is therefore minimized by the sum of these literature values divided by the counts of neurons from the BBCAv2. We studied the distribution of these ratios in the isocortex and found their average value to be 17.5% and their maximum to be 34.4%. The distribution of the ratios of GAD67 counts from literature divided by the counts of neurons from the BBCAv2 spans from 16.9% to 34.4% with a mean value at 28.5% (see S4 Fig). This indicates either that the expected 20% ratio of inhibitory neurons is incorrect in some regions of the isocortex or that the distribution of neurons from the BBCAv2 obtained at step 2 is incorrect (see Fig 1).

**4.8.2. First density estimates of the Rest population in the whole mouse brain.** By definition, the estimates of the Rest population are defined according to the proportion of inhibitory neurons that is neither PV+, SST+ nor VIP+ (see section 2.4). As we discussed in Section 4.7.3, our counts of PV, SST and VIP are reliable because of the quality of the data used to constrain them. Hence the reliability of our Rest estimates depends mostly on the GAD67+ densities and should therefore be considered with caution.

We found that the Rest population corresponds to a large proportion of inhibitory neurons in the whole mouse brain. In the isocortex, multiple literature findings show that the proportion of the Rest population should be low [58,98]. However, this is not the case in the other brain regions. For instance, the striatum is a region substantially, if not entirely, inhibitory (see review from Tepper et al. [84]). Among the inhibitory neuron population in this region, 95% of them are medium spiny neurons. This population has been shown by Zeisel et al. [69] to be expressing neither PV, nor SST nor VIP. Similarly, Zeisel et al. reported multiple neuron types belonging to the Rest population, especially in the olfactory bulb and in the brainstem. This indicates, as discussed in Section 4.4, that our equations might need to be refined in some regions of the brain, based on literature composition.

## Conclusion

The updated Blue Brain Mouse Cell Atlas pipeline estimates the numbers and densities of GAD67+, PV+, SST+ and VIP+ neurons in each region of the mouse brain according to a methodology employing assumptions and literature values. It also captures a set of brain regions in which our assumptions cause contradictions between estimates (from literature or the transfer functions) that we correct with an optimization algorithm. Our future improvements will therefore aim to reduce the errors that our analyses detected at the end of the pipeline. Future work will involve: integrating more literature points as they become available, improving our integration of the input data from the AIBS, refining our assumptions locally, and finally considering new methods for the fitting. Further analyses can study ratios and correlations of the different cell types of the brain. The workflow can be applied to any pair of reference volumes and to any genetic marker dataset from the AIBS website, which means that this pipeline can be extended to other cell types, if enough cell density estimates from literature are available to allow a fitting of mean expression data.

This project aims to involve the scientific community to contribute with open access to data, software, and tools. Our results will be released at http://bbp.epfl.ch/nexus/cell-atlas (Blue Brain Cell Atlas website interface) and the code accessible at https://github.com/BlueBrain/atlas-densities. The BBCAv2 end product is expected to be of use for many purposes. For instance, in a companion paper [57], we use our inhibitory neuron density estimations in the isocortex to estimate the densities of morpho-electrical neuron subtypes. Our model allows experimentalists to understand regional composition and permits computational neuroscientists to place defined cell types in their simulations. Furthermore, it sets the stage for further subdividing of inhibitory interneurons into more fine-grained subclasses and allows the neuroscience community to identify areas where current knowledge can be enhanced by additional constraints.

## Supporting information

**S1 Fig. Global impact of the annotation atlas on cells and neurons densities.** (A), (B), (C) Distribution of cell and neuron density values reported in literature against generated densities for similar regions, using the 3 pairs of annotations and Nissl volumes (from left to right: CCFbbp, CCFv2 from the AIBS, CCFv3 from the AIBS). When multiple literature sources are

available for the exact same region, they are both shown as a data point and are linked together. The color encodes the brain regions according to the AV, while the shapes of the points encode for cell types. The middle line delimits equal quantities, while the dashed line shows the average deviation of 2.7-fold between literature values reporting on the same region. Some subregions of the brain are not represented in CCFv3 which explains the different numbers of points. (D), (E), (F) Histogram of the brain regions in terms of neuron density values for the 3 pairs of AV and Nissl volumes (from left to right: CCFbbp, CCFv2 from the AIBS, CCFv3 from the AIBS). Each region is represented with a single-color patch of the same size. (TIF)

**S2 Fig. Algorithm 1: Cap inhibitory densities to number of neurons.** The estimated counts of GAD67+ neurons and the sum of PV+, SST+, VIP+ neurons counts are limited by the previously-computed neuron counts (step 2 of the BBCAv2 pipeline—see Fig 1) to ensure that assumptions 2 and 3 are fulfilled (see Section 2.4). Recall that **R** is the set of brain regions, $\forall$ r $\in$ **R** inversely ordered according to their depth in the region hierarchy of the AV ($\mathbf{R_o}$), the algorithm checks if the conditions $nNeu_r \geq nGAD_r$ and $nNeu_r \geq nPV_r + nSST_r + nVIP_r$ are satisfied. If not, it finds a solution which tries to match the following properties, ordered by priority: (1): Remain in range of the standard deviation of each value (confidence intervals), (2): Maintain the proportion of PV, SST and VIP within the region. If a solution exists within the confidence intervals, then the number of extra neurons (diff variable at line 19) is subtracted proportionally to the ratio of each neuron type (computed at lines 16–18). Then, if one of the neuron type estimates reaches the minimum of its confidence interval, then it is no more reduced (lines 20–22) and the remaining extra neurons are subtracted from the other neuron type estimates (lines 25, 28, 31). (TIF)

**S3 Fig. Algorithm 2: Maintain inhibitory densities coherence.** This algorithm corrects the estimated densities of PV+, SST+, VIP+ and GAD67+ neurons in the model so that assumption 4 of Section 2.4 is fulfilled. **R** is the set of brain regions, $\forall$ r $\in$ **R** inversely ordered according to their depth in the region hierarchy of the AV ($\mathbf{R_O}$), the algorithm checks if the condition $nGADr \geq nPV_r + nSST_r + nVIP_r$ is satisfied. If not, it finds a solution which tries to match the following properties, ordered by priority: (1): Remain in range of the standard deviation of each value (confidence intervals), (2): Maintain the proportion (ratios) of PV, SST and VIP. When a set of value exists so that the property (1) is fulfilled, there is a correction factor q $\in$ [0,1] which corresponds to the fraction of standard deviation needed to guarantee that the sum of the inhibitory subtypes remains under the estimated count of inhibitory neurons and that each value remains within its confidence interval. (TIF)

**S4 Fig. Explaining the isocortex inhibitory densities from BBCAv2.** (A) Coronal view of the Nissl reference atlas used for BBCAv2, showing cells of the mouse brain. Regions with high cell density appear in dark grey. The blue, orange and green lines behind the sagittal slice highlight the rapid changes of Nissl expression coming from the original Nissl experiment from Dong [18]. (B) (C) Evolution of the median expression level along the sagittal axis, in the Nissl volume realigned in Erö et al. [17], for the whole brain (B) and the isocortex (C). The rapid changes of Nissl expression that were detected in (A) are also visible in (B) and (C). (D) isocortex ratio of inhibitory neurons according to literature. The distributions show to ratios literature cell type counts in isocortex divided by the counts of neurons of the BBCAv2 in their corresponding region. The bottom and top distributions correspond to the proportion of respectively the sum of the reported values of PV, SST and VIP counts, and the reported values

of GAD67. The mean value of each distribution is shown in orange. The minimum and maximum values are indicated by the whiskers.
(TIF)

**S5 Fig. Orientations field and depth computation of the barrel cortex.** (A) Coronal slice of the AV showing the barrel cortex and its different sublayers. (B) Coronal slice of the scalar field of the barrel cortex. A weight is assigned to every voxel of the AV. These weights follow the order of crossing the isocortex by its fibers from the corpus callosum to layer 1. The borders of the barrel field are highlighted in orange. The weight assigned to the surrounding voxels of the region corresponds to their closest layer's. The weight of the voxels outside the region beyond layer 1 increases as moving away from the barrel cortex. (C) Coronal slice of the orientation field of the barrel cortex. To each voxel of the AV, a 3D direction normalized vector is computed corresponding to the main axis of the axons in the region. Colors represent the orientation vectors norm on their respective plane, black lines their projected axis. (D) Coronal slice of the depth according to pia in the barrel cortex expressed in micrometers.
(TIF)

**S6 Fig. Cortical inhibitory density distribution.** (A) Normalized distribution of cortical inhibitory neuron densities across the isocortex subregions grouped by layers. For each layer, the fitted normal distribution associated is displayed as a line on top of it. The normal distributions of L2/3 (mean $1.6 \times 10^4$, std. $3.8 \times 10^3$), L4 (mean $1.9 \times 10^4$, std. $4.1 \times 10^3$) and L5 (mean $1.8 \times 10^4$, std. $3.1 \times 10^3$) are overlapping significantly. (B) Distribution of inhibitory neuron densities according to cortical layers for each subregion of the isocortex. For most of the cortical subregions, the density is constant from L2 to L5 and drops for L6.
(TIF)

**S7 Fig. GAD67 and GAD65 ISH sagittal slices showing the dentate gyrus.** GAD67 (A) and GAD65 (B) ISH sagittal slices (images are respectively the 14th slice of experiment #75457536 and the 11th slice of experiment #79903740) of the mouse brain from the AIBS website, showing the dentate gyrus (region resembling a greater-than symbol). Somas reacting to the marker stand out from the background. The more a cell is reacting to the marker the darker it will be shown in the image. Two different populations of cells reacting to the markers can be seen in the images. First, a population of cells with large somas, and strongly reacting to both the GAD67 and GAD65 markers. Second, a dense population of cells with small somas, reacting to the GAD67 marker but not to GAD65.
(TIF)

**S8 Fig. Results of the realignment of a PV ISH coronal slice to the Nissl volume.** This figure shows coronal slices of the mouse brain from AIBS experiments of Nissl and PV (experiment id #868). ISH images from the AIBS (A) are realigned to their corresponding slice in the Nissl volume (B) using the Krepl et al.'s algorithm [56]. The registration is performed on the raw images from the AIBS as they provide more landmarks and then applied to the filtered images (C).
(TIF)

**S9 Fig. Linear fitting of marker intensity to cell density.** Scatter plots of the PV+, SST+, VIP + and GAD+ densities reported in literature (y-axis) according to the region mean intensity (x-axis). Each point represents a single literature density value. (A) The scatter plot is color-coded according to the different levels of confidence from literature data (ratio of standard deviation over mean value). The linear fit is represented with a black line. (B) Same scatter plot as (A) but the points are here color-coded according to the genetic marker expressed by the

neuron population.
(TIF)

**S1 Excel. Literature review of PV+, SST+, VIP+, and GAD67+ neuron densities in the mouse brain.** This file lists papers reporting quantification of neurons reacting to GAD67 (first sheet) and PV, SST, VIP (second sheet). For each paper and for each matching brain region of the AV, a mean density of neurons and its corresponding standard deviation is extracted. Assumptions on annotations and preprocessing steps are described in the comments' column. The age of the mice used by the papers is also indicated. The data extracted from the BBCAv2 model, used to convert reported counts or proportions into densities are stored into the last column. The last sheet contains the list of regions known containing only excitatory or inhibitory neurons and their deduced corresponding inhibitory densities of neurons according to the BBCAv2.
(XLSX)

**S2 Excel. Results of the BBCAv2 pipeline and additional analysis.** The first sheet of this file contains the final densities and standard deviation of each neuron type and for each region of the AV based on the BBCAv2 pipeline. The final column stores the volume of each region of the brain. The second sheet contains the result of the fitting of the transfer functions from region mean expression to cell density for each genetic marker (see Section 2.6). Two different ISH experiments from the AIBS were used for the GAD67 (see Section 2.4 and 3.4). The results of the fitting for each experiment and for the combined dataset are stored in different columns. The final sheet contains the densities of each neuron types for each region of the AV obtained with the BBCAv1 pipeline. The process to obtain these values is described in Section 2.1.1 and 3.4.
(XLSX)

**S1 Document. Supplementary materials.** This document contains the supplementary methods compute cell orientations and the literature review. It also contains the table of abbreviations used in this paper (Table A).
(DOCX)

## Acknowledgments

We would like to thank the Allen Institute for Brain Science for the large array of publicly available data.

Our literature review is compiling the work of many papers (see S1 Document). Any use of the results presented in this review should give the credit to the corresponding paper these results were extracted from. In particular, the major contributor of this review is the Kim et al. [7] study, and it should get a particular attention when our work is cited.

We would also like to thank K. Holm, S. Battini, A. Antonietti, H. Dictus, and F. Schürmann for comments and helpful discussions.

## Author Contributions

**Conceptualization:** Dimitri Rodarie, Csaba Verasztó, Yann Roussel.

**Data curation:** Dimitri Rodarie, Csaba Verasztó, Yann Roussel.

**Formal analysis:** Dimitri Rodarie.

**Investigation:** Dimitri Rodarie, Csaba Verasztó, Yann Roussel.

**Methodology:** Dimitri Rodarie, Csaba Verasztó, Yann Roussel.

**Project administration:** Srikanth Ramaswamy, Henry Markram, Marc-Oliver Gewaltig.

**Software:** Dimitri Rodarie, Csaba Verasztó.

**Supervision:** Henry Markram, Marc-Oliver Gewaltig.

**Validation:** Dimitri Rodarie, Csaba Verasztó, Yann Roussel, Michael Reimann, Daniel Keller, Srikanth Ramaswamy, Henry Markram, Marc-Oliver Gewaltig.

**Visualization:** Dimitri Rodarie.

**Writing – original draft:** Dimitri Rodarie, Csaba Verasztó, Yann Roussel, Michael Reimann, Daniel Keller, Srikanth Ramaswamy, Henry Markram, Marc-Oliver Gewaltig.

**Writing – review & editing:** Dimitri Rodarie, Csaba Verasztó, Yann Roussel, Michael Reimann, Daniel Keller, Srikanth Ramaswamy, Henry Markram, Marc-Oliver Gewaltig.

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
