## [Decision Letter · Decision Letter 0]

15 Apr 2022

Dear Rodarie,

First of all really sorry for the delay, we had trouble securing editors and reviewers. Apologies.

Thank you very much for submitting your manuscript "A method to estimate the cellular composition of the mouse brain from heterogeneous datasets" for consideration at PLOS Computational Biology.

As with all papers reviewed by the journal, your manuscript was reviewed by members of the editorial board and by several independent reviewers. Your paper was overall very well received. In light of the reviews (below this email), we would like to invite the resubmission of a significantly-revised version that takes into account the reviewers' comments.

We cannot make any decision about publication until we have seen the revised manuscript and your response to the reviewers' comments. Your revised manuscript is also likely to be sent to reviewers for further evaluation.

Sincerely,

Daniele Marinazzo

Deputy Editor

PLOS Computational Biology

Daniele Marinazzo

Deputy Editor

PLOS Computational Biology

Reviewer's Responses to Questions

**Comments to the Authors:**

Reviewer #1: Rodarie et al has estimated numbers and densities of cells expressing GAD67, as well as subsets of these neurons expressing PV, SST and VIP. In order to estimate densities in areas where information is currently lacking, they have combined analyses of publicly available ISH data from the Allen Institute with numbers available in the literature. They use these data to create 3D representations of the specific neuron types across the brain, which they present in an interactive web-based viewer. This presentation through the viewer is impressive and I believe of broad interest in the community (unfortunately I cannot currently find the data presented in this paper, but assume it is going to be released upon publication of the article). While I am overall supportive of the authors aims, I have several concerns about the methodology and conclusions drawn.

MAJOR:

- Cell density estimations:

1. The authors use a binary thresholding method on ISH images and then attempt to use the intensity values and literature data to predict cell densities in regions where no literature is available. The core assumption, as stated in section 2.1, is that there is "a correlation between genetic marker expression and cell type density gathered from the literature and the previously aligned and filtered ISH datasets". However, this assumption is shown by the authors not to be true, as evidenced by the poor performance of the transfer function. The authors show that there is no significant improvement when averaging two datasets for a marker. The qualitative validation (Fig. 10) further shows major differences between the real and the predicted sagittal data. Given all of this, I find it curious that the authors do not choose another approach for estimating cell densities from the AIBS images. They mention this opportunity in the discussion, but only as a future direction. I think it should be considered for the current paper, at least for the GAD67 dataset where literature data are so scarce, but preferably for all the markers. My comments below will highlight why this is of importance for the conclusions of the study.

2. The most important consequence of the above point is that the resulting density estimates are heavily biased by the method used to compute them. For the three interneuron subtypes for which data are presented in this study, brain-wide estimates are available from the literature (from Kim et al). In contrast, cell density estimates for all interneurons (assessed through GAD67 expression) is scarcely available. The main contribution then - in terms of new data - are the estimates of GAD67 and (by subtraction of the specific types above) the proportion of interneurons not expressing PV, SST or VIP. The finding that such a large proportion of GAD67 neurons are not expressing any of these proteins is highly unexpected, as these cell types are typically considered to make up most of the interneuronal population across the brain (Rudy, 2010). However, given that the literature estimates for the GAD67 population covers so few of brain regions compared to the other three cell types, these numbers are in >80 % of cases based on the intensity analysis and transfer function, which the authors quite clearly demonstrates does not correspond well with empirical data from the literature. A more refined method for extracting cell densities from the AIBS images (as mentioned in the comment above) would be needed substantiate the claim that such a large proportion of GAD67 neurons to not express PV, SST or VIP. If it is indeed true, this is a very interesting finding that should be discussed further. However, without further confirmation I am inclined to believe that the incongruence between the PV+SST+VIP numbers and total GAD67 numbers are rather a byproduct of using very different methods to compute them.

3. While the integration of the Kim et al. data in the author’s Cell Atlas is powerful, and the other estimations might be useful as a first approximation for computational models, the authors are making claims about these data that are not justified given the limitations and biases of their approach. For example, the claim in the introduction that "these refinements will help neuroscientists to get a better understanding at cellular composition" and the use of terms like “unbiased” and “coherent estimates”.

- Choice of reference atlas and registration of data to it:

1. In the introduction, the authors state that "alignment of multiple datasets to a common coordinate system presents a significant source of uncertainty that we measured and addressed with deep learning techniques". I believe that the authors with “the source of uncertainty that we measured” are referring to their selection of brain reference atlas based on the resulting cell density distribution (Section 2.2). However, in this section they are measuring the fit of different combinations of the AIBS nissl and annotation volumes. They do not address the uncertainty in alignment of multiple datasets (e.g. the ISH data used in their analysis). A notable source of uncertainty is that histological sections typically deviate from the standard plane, this is evident also in the AIBS ISH data (particularly in the somatostatin data used in the current study). It is not clear whether the authors' approach account for this. The authors should measure (or at least discuss) the uncertainty in their registration of the ISH data. They should also show images of the registration results for the ISH data so that the reader can qualitatively assess the fit.

2. Section 2.2 gives a summary of the different Nissl and annotation versions from the Allen Institute, which provides important background for the reader to understand the methodological decisions of the authors. However, I find this part to be inaccurate or wrong in several places, for example:

a. The authors do not mention the different versions of the annotation volume for CCFv3, and do not specify which one was used in their own analysis for selecting the reference atlas (Section 2.2).

b. According to the technical white papers from the AIBS, CCFv1 was published in 2005 and contained ~200 delineations. Clearly it is not this version that the authors have used in their analysis of the ISH data, since the number of brain regions in their AV is 861.

c. It is misleading to say that subregions are "merged" in the AV3, as all annotations were re-drawn in 3D for this version of the CCF

The authors should sort out these points and should also stay as close as possible to the terminology and versioning scheme used by the Allen Institute; currently there is some mixing of terms like "AV" and "AMBA".

OTHER COMMENTS:

- The methodology for the literature search is very scarcely described, providing no details on e.g. how, when, or through which search engine was performed nor of the search strings employed.

- It is good that the supplementary file giving the data and details from the survey is included, but the file needs revision for clarity and overall organization. There are also some inconsistency with the information provided in the text, e.g. the main text says there are 29 PV papers, but supplementary file only contains 23 references in total for PV, SST and VIP. In the result it is stated that they “integrated values from 56 different literature sources”. In the discussion, they state that they perform a review of quantitative stereological studies, but several of the sources are not using stereology as the cell counting method.

- I believe all references from the literature search should be cited in the main text, as these papers have provided important data for the current study that should receive proper citation.

- I find the introduction and discussion to be too sparse on references to previous work. There are several studies available on brain-wide mapping of various cell types. The authors should discuss how their approach differ from and compares to other approaches.

- Figure 1 shows the steps of the BBCAv2 workflow with input (Data) and output (Results) of each step in different rows. It is stated in the text that "each step of this pipeline builds on the results of the previous step", but this is not entirely clear from the figure because the wording is not consistent across panels. For example, it also seems that the three data points in step 4 correspond to the output, respectively, from steps 1-3, but the points are phrased different from the results in the individual panels.

- The variable use of terms like “thresholded”, “filtered”, and “binarized” images in the methods section is somewhat confusing

Reviewer #2: This is a monumental effort to generate an cellular atlas of the mouse brain. The paper is methodological, improves a previous work considerably, is rigorously conducted and well written. It surely sets a step forward in the generation of atlases for cellular reconstruction of brain models.

Minor .

The paper could gain in clarity by making some points more explicit:

1) Spell out clearly, possibly in the introduction, that the atlas is itself a model. The atlas is not just a collection of elements, but it also requires an interpolation from available data

2) Explain clearly what is not available from data and requires an advanced atlas reconstruction.

3) How was the literature search carried out? It is not clear if all the paper used for model correction (e.g., table 1) are reported in the references. If not, a complete list could be added in the supplementary material

4) A glossary would be useful, since there are several non-standard abbreviations that are not always easy to track. E.g., ISH, AV, AIBS, CCF, BBCA, etc.

5) Fig.5: x-axis is marker intensity? please specify.

6) Fig.6: a higher resolution would be useful

7) The man result is the comparison of v1 to v2 of the BBCA. This should be discussed more thoroughly. In general, physiological and anatomical implications are somehow diminished compared to the methodological part. A better discussion of results is needed. For example, table 4 shows interesting predictions on the inhibitory neuron populations ion different brain regions. A quantitative mapping to known neuronal populations is possible in several cases and should be discussed. E.g., for cerebellum inh/neu=13% seems unusually low but it matches the presence of a huge population of excitatory granule cells. In striatum, inh/neu=81.4% is unusually high and matches the predominance of inhibitory cells. Etc .On the same line, an homologous of Fig.8 should be done also for cerebellum and basal ganglia.

8) Fig. 9 should be better commented. In 9c, the increase of literature data should not increase R2 ?

Reviewer #3: Overall:

In the various worldwide efforts to extend models of neural tissue to larger parts or even the entire brain, a catalog of cell densities for the major celltypes in the brain is one of the essential ingredients. The present research is a significant update to a previously published cell-type catalog for the mouse brain. It has the potential to become a reference resource for this type of data, and to motivate new research that fills gaps in experimental data or refines the model.

In comparison with the previous iteration, the current model returns cell-density values that are closer to literature values

while taking care of maintaining consistency across length-scales. It also incorporates more inhibitory cell types, thereby relying quite heavily on the data of Kim et al. 2017.

The most difficult parts to achieve good quality estimates turn out to be:

- getting cortical layer borders in the correct position, a slight misalignment has huge consequences.

- estimating the total number of GABAergic neurons, for which the literature data is very sparse.

The manuscript starts with a discussion on the space in which the data is presented, thereby choosing among

different iterations and optizations of the Allen Mouse Brain Atlas.

The latest incarnation of that atlas, "CCF version 3" is widely used in the community to register a wide variety of data to, including 10,000 fully reconstructed projection neurons, and I am a bit disappointed that the authors did not find a way to map their data onto this template, using an older version of the atlas instead.

The authors motivate their choice by pointing out that the Dong NISSL sections, needed for the cell density pipeline, have not been registered well to CCF3, especially when looking at the borders of cortical layers. I do accept this choice for the scope of the current manuscript, but hope that they will follow up on the plan outlined in the discussion to re-register the NISSL sections to CCF3 in such a way that the borders of cortical layers match those of the annotation.

As for the cortical layer border alignment issue, have you tried to work with an exclusion zone, i.e. not take into account data that is very close to a cortical layer border?

Another concern that I have is that the authors combine literature values from many sources, but in absolute numbers I think one of these references stands out from the crowd with head and shoulders, namely Kim et al. 2017. I have some experience with other databases that combine literature values, and this can create problems with citation scores: if all subsequent studies would cite the current study for cell densities and not the extensive work of Kim et al. 2017, then the latter author would not get sufficient credit for his work.

This can be taken care of in a citation policy that points out the terms of use of the database and which studies must be cited depending on how the database is used. Please consider adding such a policy to the manuscript and website.

Overall the manuscript is well written and the figures are of good quality. I highly appreciate the work, the various steps of the pipeline are well motivated and state of the art, and the detailed comments below should be seen in this context.

Detailed comments:

Line 166:

are not sufficient to estimate the density of all inhibitory subtypes in each voxel of the mouse brain

=> mention voxe size, 10, 25 or 100 microns?

Line 169:

Since several AV and Nissl volumes were recently released [33],

=> I don't think [33] introduced a new Nissl volume, but rather an averaged STP dataset.

Near line 172, Figure 1, row 3:

Compute a number of neuron expressing...

=> compute the number of neurons expressing...

and same Fig row 4:

Estimates counts

=> Estimated counts

Near line 232, Figure 2 part D:

Normalized Neuron Densities

=> Normalized with respect to what, and why choose a different axis than in part C?

Line 267:

CCFv3 version (Nissl2 + AMBA3) from the AIBS [33].

=> Previously you used the term AV3, now AMBA3, you probably did not apply your change throughout the document.

Line 268:

For each pair of reference atlases, ...

=> Confusing, an atlas is a combination of one or more imaging modalities (space) and one or more delineations (annotation).

I think what you mean is "For each variant of the reference atlas, ..."

Line 281:

(see S1 Fig.)

Missing Figure number

Line 309:

In brief, this algorithm constructs a displacement field by finding pixel by pixel correspondences between an image and a reference. Applying these displacement fields as geometric transformations, images are aligned to the same reference and then converted to a 3D volume at the same resolution using downsampling.

=> I suggest to leave out this paragraph, since you explain the procedure again in much clearer terms in the next paragraph.

Line 322:

For each dataset, every coronal section is automatically realigned to the corresponding anatomical section of the Nissl-stained mouse brain using the Krepl algorithm [31].

=> Knowing that this is not the subject of this manuscript: how does the Krepl algorithm find which is the best section in the Nissl volume that corresponds to a coronal section from an ISH dataset? Also in a quick scan of the Krepl paper I could not find this.

Is the procedure 'just' a 2d-2d registration, assuming that the Allen metadata is correct insofar slice pairing is concerned?

Line 429:

For some regions such as the CA1 field of the hippocampus, we observe a large variability of

the literature estimates (see Fig. 4B): spanning from 317 GAD67+ cells per mm computed from

431 Han et al. [54] to 7166 GAD67+ cells per mm according to Jinno et al. [55]

=> You do list factors that could contribute to the large spread in these estimates, but it is still unsatisfactory to not know which number in this particular example is the wrong one and why. It is worthwhile to contact the authors of the conflicting numbers to clarify the cause, there can be confusion about units (in the Han et al. paper they seem to mix neurons/mm2 with 'number of neurons').

This may also help with the poor fit in Figure 5 for the GAD/Rest subplot.

Line 508:

Use of the term 'orig' for your estimates: I first thought this was inspired by the 'original ISH data of the Allen Institute', but when reading on it turns out that you are going to correct this with constraints and want to distinguish 'original' from 'corrected'.

I would avoid using the word 'orig' as a symbol for your model predictions, rather use a greek letter.

In the text you could replace 'original' with unconstrained.

Paragraph 2.6

You do not make sufficiently clear why you first "count every cell reacting to the markers." in the ISH data, then convert this to an all-or none voxelized image, and then use the transfer functions to convert back to cell count.

Why can't the counted cells in the ISH data be used directly?

Figure 6, end of caption

"The variation of the distribution of neurons is following the original distribution of Nissl expression. The Nissl expression drops significantly after the blue line which leads to a similar decrease of cell counts in the resulting BBCAv2."

=> This is a remark that requires much more attention.

When looking at Fig 6C there are many places where the NISSL staining is darker/lighter,

so why is the level where you show the blue line so special?

And if the NISSL data is used for counting cell bodies, why does it matter whether the staining is darker/lighter?

Are you saying here that the NISSL data is wrong and that your model is also wrong?

Probably not but that is how it reads.

Line 609:

The result is an updated mouse cell atlas which provides a 3D position and a type label for each cell within the CCFbbp AV (see Fig. 6CD).

=> The result is a table with predicted densities and standard deviations for each cell class in all regons of the AV.

A realization of this model with randomly placed cells within each brain region was computed and published as the Blue Brain Mouse Cell Atlas.

Please choose wording such that it becomes clear that the BB Cell Atlas represents a single realization of your model.

Line 614:

The link https://bbpgitlab.epfl.ch/nse/atlas-building-tools

points to an inaccessible location: "Unable to connect"

Line 652 and 654

"We predict that 25% of the neurons of the mouse’s isocortex to be inhibitory"

=> grammar error, predict that ... is or predict ... to be ...

Line 679-683

It is not immediately clear *why* you studied the density profiles wrt. pia so extensively, resulting in Fig. 8CD and an extended methods section. I think it is better to start by saying that Meyer et al. studied this profile and that you wanted to see whether the model is consistent with that.

Line 748:

Additionally, the granule layer of the dentate gyrus is clearly visible on the original experiment

but not in the BBCAv2 brain slice (center black arrow).

=> This prominent difference is not explained or discussed in the text. This layer is distinct part of the Annotation Volume, so how can it be that the model does not predict a correct value for it?

Line 780:

"It is therefore usually underestimating the densities of each neuron type according to literature, since the neuron and inhibitory neuron densities act as maximum constraints."

From Fig. 10E, I don't see that the new model underestimates densities.

And, in the cortex the inhibitory neuron density is 25% according to the model vs. 20% in Literature.

That is an overestimation, not underestimation.

Be more specific with the statements so that this confusion is avoided.

Line 851:

The marker datasets, despite having been processed by the AIBS, are still subject to artefacts

and noise which will artificially raise our cell counts estimates (see Fig. 3A).

=> I see artefacs in 3A, but not in 3C. It could also lower the estimates, isn't it?

Line 852:

To counter these effects, Erö et al. [14] used the Nrn1 marker, as it is expressed in excitatory neurons, to limit the expression of GAD67.

=> How can the expression in one experiment be limited by the expression of another substance in another animal? This needs more clarification. Is it because you assume that the GAD67 expression density must not be higher than some preset fraction of the Nrn1 expression density?

**Have the authors made all data and (if applicable) computational code underlying the findings in their manuscript fully available?**

Reviewer #1: Yes

Reviewer #2: Yes

Reviewer #3: **No: **The link to their git repository "" ext-link-type="uri" xlink:type="simple">https://bbpgitlab.epfl.ch/nse/atlas-building-tools" is not public.

PLOS authors have the option to publish the peer review history of their article (what does this mean?). If published, this will include your full peer review and any attached files.

Reviewer #1: No

Reviewer #2: No

Reviewer #3: **Yes: **Rembrandt Bakker

Figure Files:

Data Requirements:

Reproducibility:

To enhance the reproducibility of your results, we recommend that you deposit your laboratory protocols in protocols.io, where a protocol can be assigned its own identifier (DOI) such that it can be cited independently in the future. Additionally, PLOS ONE offers an option to publish peer-reviewed clinical study protocols. Read more information on sharing protocols at https://plos.org/protocols?utm_medium=editorial-emailutm_source=authorlettersutm_campaign=protocols

---

## [Decision Letter · Decision Letter 1]

6 Oct 2022

Dear Rodarie,

Thank you very much for submitting your manuscript "A method to estimate the cellular composition of the mouse brain from heterogeneous datasets" for consideration at PLOS Computational Biology. As with all papers reviewed by the journal, your manuscript was reviewed by members of the editorial board and by several independent reviewers. The reviewers appreciated the attention to an important topic. Based on the reviews, we are likely to accept this manuscript for publication, providing that you modify the manuscript according to the review recommendations.

Sincerely,

Daniele Marinazzo

Section Editor

PLOS Computational Biology

Daniele Marinazzo

Section Editor

PLOS Computational Biology

Reviewer's Responses to Questions

**Comments to the Authors:**

Reviewer #1: Overall, I think the authors in this revision have provided important clarifications for readers to understand their approach, as well as interesting new observations to support their data. I do, however, have some remaining comments that should be addressed:

- I think the revision help clarify many of the limitations and advantages of the authors approach related to the transfer function and density estimates, and understand the authors’ point that the literature variability presents a difficulty for the fit better after this revision. However, the authors have not really addressed my concern that the density estimates may be biased by the method used to compute them. This is in a way parallel to how estimates in the literature vary a lot, probably because such different methods are used across studies. Thus, I still believe the authors should discuss how their GAD to subtypes ratios could be affected by the fact that brain-wide data from a single study were available for all three subtypes but not the GAD cells.

- I appreciate the added observations in section 3.4. and the new S7 Fig, and think it adds balance to the qualitative comparison. However, Figure S7 does not say which image represents which staining. It is also not clear to me what the authors are referring to by the "other population of cells", smaller in diameter and lower GAD67 population. It would be useful if they could add labels and pointers in the figure to more clearly indicate their observations. Lastly it should be made clear which Allen brain atlas experiments and slices are depicted in this figure, and they should be cited in the main text with links so that the reader can easily go in and look at the high resolution images.

- Similarly, the sagittal section from the AIBS shown in figure 10 should also be cited appropriately with experiment ID so readers can find it.

- I think the language should be critically reviewed across the manuscript, and especially within the newly added text, where there are quite a number of typos and poorly formulated sentences.

**Have the authors made all data and (if applicable) computational code underlying the findings in their manuscript fully available?**

Reviewer #1: Yes

PLOS authors have the option to publish the peer review history of their article (what does this mean?). If published, this will include your full peer review and any attached files.

Reviewer #1: **Yes: **Ingvild Elise Bjerke

Figure Files:

Data Requirements:

Reproducibility:

To enhance the reproducibility of your results, we recommend that you deposit your laboratory protocols in protocols.io, where a protocol can be assigned its own identifier (DOI) such that it can be cited independently in the future. Additionally, PLOS ONE offers an option to publish peer-reviewed clinical study protocols. Read more information on sharing protocols at https://plos.org/protocols?utm_medium=editorial-emailutm_source=authorlettersutm_campaign=protocols

References:

---

## [Editor Report · Decision Letter 2]

15 Nov 2022

Dear Rodarie,

We are pleased to inform you that your manuscript 'A method to estimate the cellular composition of the mouse brain from heterogeneous datasets' has been provisionally accepted for publication in PLOS Computational Biology.

Best regards,

Daniele Marinazzo

Section Editor

PLOS Computational Biology

Daniele Marinazzo

Section Editor

PLOS Computational Biology

---

## [Editor Report · Acceptance letter]

13 Dec 2022

PCOMPBIOL-D-22-00098R2 

A method to estimate the cellular composition of the mouse brain from heterogeneous datasets

Dear Dr Rodarie,

I am pleased to inform you that your manuscript has been formally accepted for publication in PLOS Computational Biology. Your manuscript is now with our production department and you will be notified of the publication date in due course.

With kind regards,

Zsofi Zombor
